# A comprehensive thermodynamic model for RNA binding by the *Saccharomyces cerevisiae* Pumilio protein PUF4

Christoph Sadée [1,10], Lauren D. Hagler[1,10], Winston R. Becker [2], Inga Jarmoskaite [1,3], Pavanapuresan P. Vaidyanathan[1,8], Sarah K. Denny[2,9], William J. Greenleaf [3,4,5] & Daniel Herschlag [1,6,7] ✉

Genomic methods have been valuable for identifying RNA-binding proteins (RBPs) and the genes, pathways, and processes they regulate. Nevertheless, standard motif descriptions cannot be used to predict all RNA targets or test quantitative models for cellular interactions and regulation. We present a complete thermodynamic model for RNA binding to the *S. cerevisiae* Pumilio protein PUF4 derived from direct binding data for 6180 RNAs measured using the RNA on a massively parallel array (RNA-MaP) platform. The PUF4 model is highly similar to that of the related RBPs, human PUM2 and PUM1, with one marked exception: a single favorable site of base flipping for PUF4, such that PUF4 preferentially binds to a non-contiguous series of residues. These results are foundational for developing and testing cellular models of RNA-RBP interactions and function, for engineering RBPs, for understanding the biophysical nature of RBP binding and the evolutionary landscape of RNAs and RBPs.

RNAs and their interactions are integral to the regulation of gene expression. With the realization that RNA binding proteins (RBPs) bind many related RNAs to coordinate their function[1–4] and that at least one in 20 proteins in the genome bind RNA[5–8], several important challenges have emerged: (i) to identify the RNA targets of RBPs; (ii) to determine the functional consequences of these interactions; (iii) to determine the biophysical basis of RBP binding and specificity; (iv) to trace the evolution of RBPs and their interactions; and (v) to develop rules for engineering RBP affinity and specificity.

Following the recognition in the early 2000s that individual RBPs bind large sets of related RNAs, a major goal has been to identify the RNA binding partners for the many cellular RBPs[4,9–12]. Subsequent studies have helped to elucidate the roles of RNA–protein interactions in cellular processes[13–15]. As genomic-scale methods have become more and more sensitive, RBP target lists have grown in length[8,16–18].

However, RNAs have a range of affinities, and RBPs exhibit robust non-specific binding. It is unclear where, or even if, one can draw a clean cutoff between binders and non-binders, and, importantly, it is unlikely that all RNAs that bind a particular RBP are equally affected by that RBP and by changes in its levels, modifications, and localization. Most simply, RNAs with higher and lower RBP occupancies are expected to be more and less functionally affected by those RBPs, respectively, all else being equal. Thus, predictive models of RNA and RBP function must start from predictions of occupancies.

A zeroth-order model is that occupancy is simply a reflection of RNA-RBP thermodynamics–i.e., the intrinsic affinities for each sequence and the accessibility of that sequence. Genomic-scale RNA-RBP cross-linking (CLIP) experiments can in principle be used to test this model in cells. However, there are limitations to the information one can extract from genomic-wide studies. The number of

[1]Department of Biochemistry, Stanford University School of Medicine, Stanford, CA, USA. [2]Biophysics Program, Stanford University School of Medicine, Stanford, CA, USA. [3]Department of Genetics, Stanford University School of Medicine, Stanford, CA, USA. [4]Department of Applied Physics, Stanford University, Stanford, CA, USA. [5]Chan Zuckerberg Biohub, San Francisco, CA, USA. [6]Department of Chemical Engineering, Stanford University, Stanford, CA, USA. [7]ChEM-H Institute, Stanford University, Stanford, CA, USA. [8]Present address: Protillion Biosciences, Burlingame, CA, USA. [9]Present address: Scribe Therapeutics, Alameda, CA, USA. [10]These authors contributed equally: Christoph Sadée, Lauren D. Hagler. ✉e-mail: herschla@stanford.edu

sequencing reads from genome-wide eCLIP is quantitative, but it is subject to distortions at each of several experimental steps and also subject to statistical limitations from low sequence read depth. As a result, the number of reads at a particular RNA site may not faithfully reflect quantitative RBP occupancies in the cell[17]. Testing this zeroth-order model, determining when it holds and where it breaks down, and uncovering what cellular factors and features are responsible for its breakdown requires starting with comprehensive binding models to predict relative occupancies. Such simple quantitative models are most critical when attempting to disentangle complex multi-variate systems such as molecular interactions and outcomes in cells[19].

Pumilio proteins have eight pseudo repeats each recognizing one residue (Fig. 1b)[20,21]. Intriguingly, human PUM1/2[22–25] and yeast PUF4[25] have highly similar RNA-contacting amino acids (Fig. 1c) and binding motifs and logos that are similar (Fig. 1a); yet the identified RNA targets are considerably different[9,10,26–32]. These observations are fascinating from both evolutionary and biophysical perspectives. Evolutionarily, the similar binding specificities likely allowed the exchange of >100 RNA targets between *S. cerevisiae* PUF3 (an ortholog of PUM1/2) and PUF4 during fungal evolution to alter their regulation[28]. Biophysically,

determining the altered sequence specificity provides a foundation for deciphering the molecular origins for these changes.

We previously derived a complete thermodynamic model for human Pumilio proteins, PUM2 and PUM1 (referred to together as PUM1/2, Fig. 1a) binding to all RNAs. PUM1/2 have identical specificities[33] and are described by the same mathematical model contained 59 terms, with a single constant affinity offset[34]. The binding to an engineered variant of PUM1 with a single mutation within one of its Pumilio modules was quantitatively accounted for by varying just one of the 59 terms, one representing specificity within the mutated module[34,35]. Here, we derive an analogous mathematical model for the yeast PUF4 targets that provides similar high accuracy, despite the greater complexity of its motif descriptions[9,10,26,27,36]. The PUF4 thermodynamic model is based on direct high-throughput binding data for 6180 RNAs measured with the RNA on a massively parallel array (RNA-MaP) platform[37,38]. The resulting model, consisting of 56 terms, is highly similar to that for the related RBPs, PUM1/2, with one marked exception. The difference in RNA binding to PUF4 *versus* PUM1/2 arises from a single favorable flipping term for PUF4, such that PUF4 binds preferentially to a non-contiguous series of residues. All other flips and

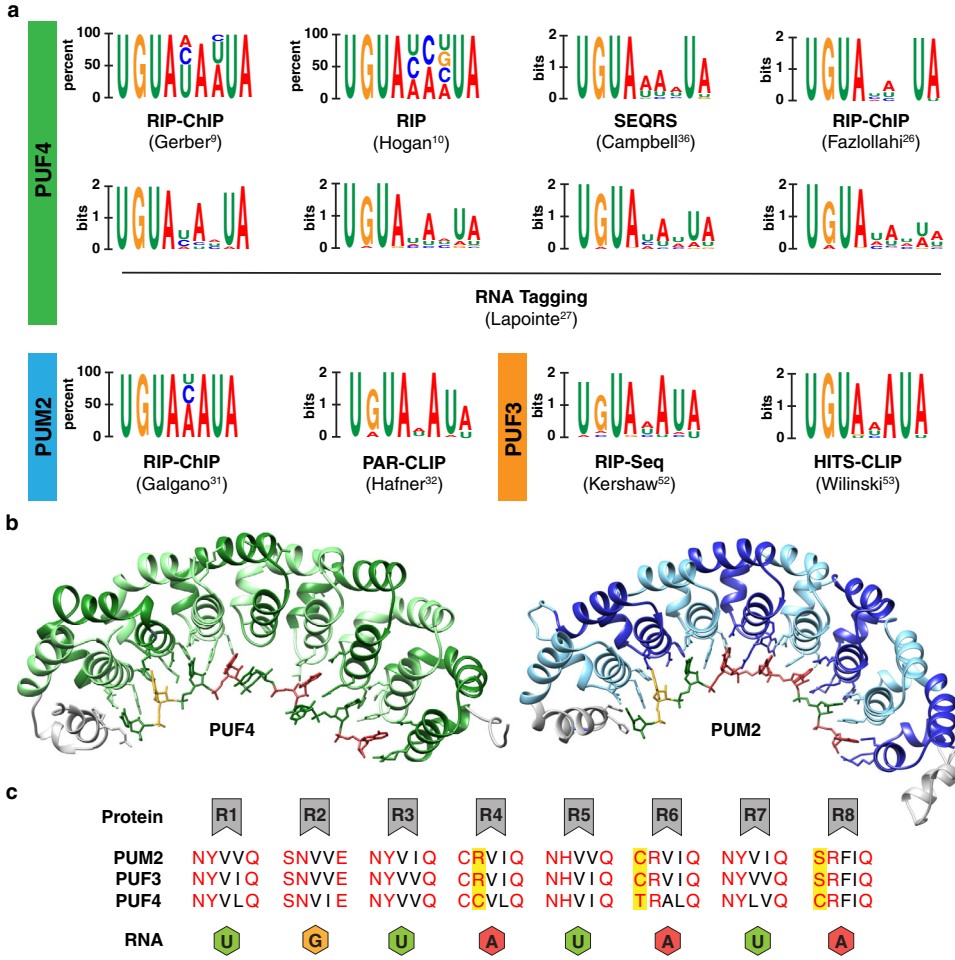

**Fig. 1 | Comparison of PUF4, PUM2, and PUF3 motifs, structure, and sequence.**
**a** Literature PUF4 sequence motifs compared to a subset of those for PUM2 and PUF3. The experimental method for each representation is shown with the corresponding refs. 9, 10, 26, 27, 31, 32, 36, 54, 55 (RIP: RNA precipitation; RIP-Chip: RNA-binding immunoprecipitation microarray profiling; SEQRS: high-throughput sequencing and sequence specificity landscapes; PAR-CLIP: photoactivatable ribonucleoside-enhanced cross-linking and immunoprecipitation; HITS-CLIP: cross-linking and immunoprecipitation coupled with high-throughput sequencing). A full list of PUM2 and PUF3 motifs is given in ref. 34. **b** Crystal structures of

the RNA-binding domain of PUF4 bound to UGUAUAUUA (left, PDB: 3BX2)[47] and human PUM2 bound UGUAAAUA (right, PDB: 3Q0Q)[22]. **c** PUM2, PUF3, and PUF4 repeat residues involved in base-specific interactions (see also Supplementary Fig. 1). For simplicity, the eight binding sites (R1-R8) are numbered in the 5′ to 3′ order of bound consensus RNA (5′-UGUAUAUA-3′), the reverse order of the protein primary sequence and opposite to literature convention. Red residues make specific contacts with RNA bases. Yellow boxed residues indicate differences between PUF4 and PUM2/PUF3. The green, orange, and red RNA bases correspond to the base favored at each position of PUM2/PUF3 based on the consensus motifs.

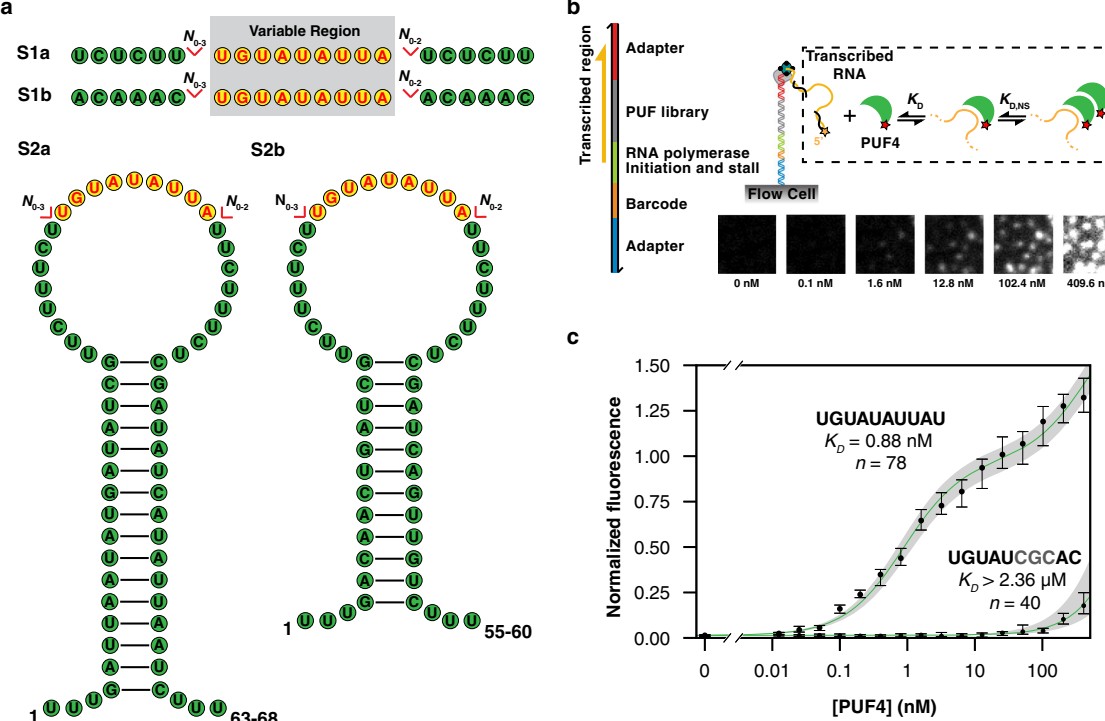

**Fig. 2 | Quantitative, high-throughput measurements of RNA binding to PUF4.**
**a** Designed RNA library to probe PUF4 binding specificity. A variable region (yellow bases) is embedded into each of four scaffolds (green bases) S1a, S1b, S2a, and S2b. Variations are introduced with reference to PUF4 and other Pumilio protein consensus sequences (PUM2 and PUF5), including one to four base mutations, one to five insertions at each consensus position, and flanking insertions ($N = 0-3$ nt). Additional sequence information is shown in Supplementary Fig. 2a. **b** Schematic representation of RNA-MaP procedure[37], including representative fluorescence images for Cy3B-PUF4 binding to a small region of a sequencing chip. **c** Representative binding curves from RNA-MaP of two RNA variants (UGUAUAUUAU in S1a scaffold and UGUAUCGCAC in S1b scaffold). Black circles

indicate median fluorescence at each protein concentration for all clusters of the respective variants normalized to the background fluorescence of the RNA channel. The number of replicate clusters is denoted by $n$. Error bars represent 95% confidence intervals (CI) across the clusters determined by bootstrap analysis. The green lines indicate the fits to the binding model. The grey shaded area indicates the 95% CI of the fit ($K_D$ (consensus) = 0.88 nM, $CI_{95\%}$ = (7.86 nM; 1.03 nM)); $K_D$ (mutant) > 2.4 μM, corresponding to the upper limit for binding affinities that could be confidently distinguished from background). For high-affinity binders an additional increase in fluorescence is observed at high protein concentrations that is accounted for by a non-specific binding term for PUF4 binding to the RNA-PUF4 complex (see ref. 34 and "Methods"). The x-axis is logarithmic.

non-contiguous binding modes are unfavorable for PUF4 as they are for PUM1/2, and the other 52 terms in the PUF4 and PUM1/2 binding models are highly similar, indicating that binding to each of the individual Pumilio repeats is the same or nearly the same. These results are foundational for testing cellular models of RNA-PUF4 interactions and function, for engineering RBPs, and for understanding the biophysical nature of Pumilio binding and the evolutionary landscape of RNAs and Pumilio proteins.

## Results

### RNA-MaP to measure the binding affinities of PUF4 to 6180 RNAs

To derive a thermodynamic PUF4-binding model, we followed the approach we used previously with the related human Pumilio proteins, PUM1/2, designing a series of RNA sequences that vary relative to the previously-identified consensus sequence (Fig. 1a)[9]. In this case, the library was designed, rather than randomized, to ensure systematic variation relative to the consensus sequence, while also allowing exploration further into sequence space without sacrificing large amounts of the library to non-binders (Fig. 2a). Other library approaches allow more sequences to be explored but sacrifice thermodynamic rigor. Also, the models from RNA-MaP, while not generated based on all possible sequences, are quantitative and predictive and can be expanded or modified if confronted by new affinities that, by rigorous statistical treatment, are not predicted by the current model.

In our library, we included mutations (1–4 nt) to probe sequence specificity and insertions (1–5 nt) to assay potential non-contiguous binding sites. Additionally, we varied the immediate flanking residues (0–3 nt) to assess possible extended binding sites (Fig. 2a and Supplementary Fig. 2a). The library included the sequence variants used to develop the PUM1/2 binding models[34] and variants from the consensus of PUF5 (an *S. cerevisiae* Pumilio protein related to PUF4)[39] to provide a range of related but different sequence variants (see Source data for full library). The RNA sequence variants were embedded in four different scaffolds to control for RNA structures that could form from interactions with particular sequence variants or flanking sequences (Fig. 2a).

We used RNA-MaP to measure PUF4 equilibrium dissociation constants directly for the entire library (Fig. 2b and Source data). We began with a DNA library encoding our RNA variants; the DNA library was sequenced on an Illumina MiSeq flow cell, followed by in situ transcription on a custom-built imaging and fluidics setup[37,40,41]. RNA transcripts were immobilized by stalling *E. coli* RNA polymerase at the end of the DNA template, and RNA–protein association was measured by equilibrating the RNA with increasing concentrations of fluorescently-labeled PUF4 and by imaging each cluster (comprising ~1000 copies of that RNA variant)[37]. The resulting binding curves were used to obtain the dissociation constant ($K_D$) and the corresponding free energy of binding, $\Delta G$ (=$RT\ln K_D$), of the protein for each RNA variant, as described in the "Methods". Figure 2c shows representative binding curves for a tight and weak binding RNA, with the tight binder

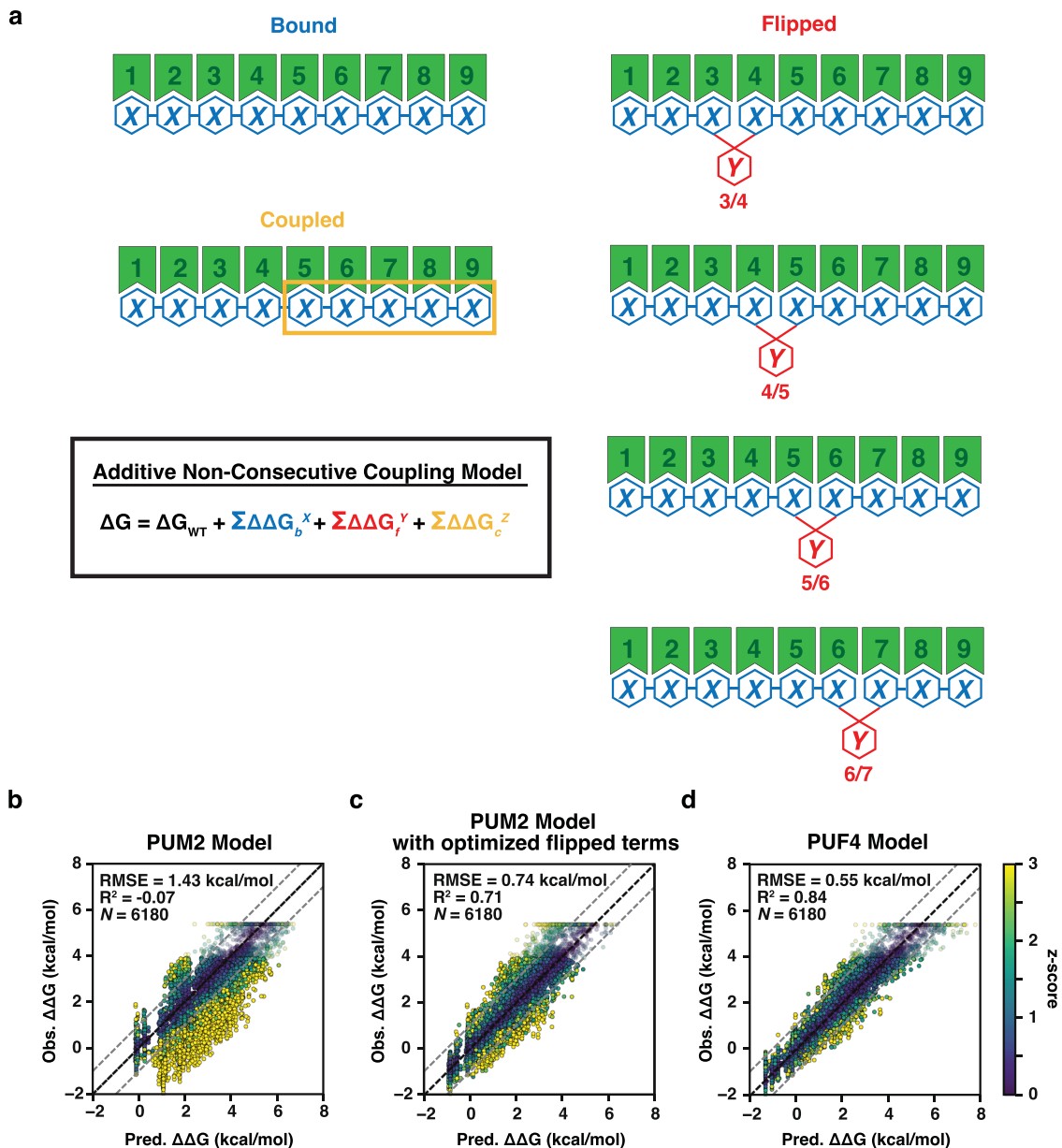

**Fig. 3 | Predictive thermodynamic models for RNA binding by Pumilio proteins.** **a** Generalized model RNA binding by an RBP, with terms for PUM2[34]. Each blue term (bound, $\Delta\Delta G_b{}^x$) represents an additive term for binding in a particular site; red terms (flipped, $\Delta\Delta G_f{}^y$) represent residues not engaged in direct binding interactions, which allows non-contiguous binding; and orange terms (coupled, $\Delta\Delta G_c{}^z$) represent higher-order interaction terms. Many more higher-order (coupling) terms are possible for a particular RBP; the terms shown were deemed necessary to quantitatively account for the binding of thousands of RNAs to PUM2[34]. **b** PUF4-binding data fit to a 59-term additive non-consecutive model using the PUM2 parameters determined in ref. 34. **c** PUF4 binding data fit to the same model using the PUM2 bound parameters but with the flipped parameters allowed to vary to optimize the fit (minimizing the sum of squared errors of Obs. $\Delta\Delta G$ − Pred. $\Delta\Delta G$). **d** PUF4 binding data fit to the additive non-consecutive model where all parameters were allowed to vary to give the best fit. Additive parameters were initialized based on single mutant penalties (see Fig. 4) and allowed to vary within 0.75 kcal/mol or their error bounds if those exceeded 0.75 kcal/mol. For **b** through **d** the x-axis is the predicted affinities based on the model and the y-axis is the observed affinities measured by RNA-MaP. Points are colored based on the z-score, or the absolute difference in predicted and observed affinity divided by the observed uncertainty, as described in the "Methods" capped at $z = 3$ for visualization. RMSE and $R^2$ were used to determine the goodness of fit to x = y (black dashed line) from a linear regression (see also "Methods").

corresponding to the PUF4 consensus sequence. As observed previously with PUM2, there is a second, non-specific binding component that represents binding of the protein to the RNA–protein complex, and these data were fit as previously done for PUM2 (ref. 34 and "Methods").

We assessed the binding of 15,272 distinct RNAs. Of these, affinities were obtained for 6180 RNAs which had measurable affinities (≤2.3 μM) and ≥5 technical replicates to provide high affinity and robust statistics. Prior work demonstrated precise measurements

within ~0.28 kcal/mol with this number of replicates, with the accuracy increasing with more replicates and decreasing with weaker binding, and our results mirrored these prior findings (Supplementary Fig. 2c; ref. 34). Equilibrium binding was measured for two salt conditions (low and high; see "Methods"). Fully independent datasets measured on two chips with high salt (2 mM MgCl$_2$, 100 mM KOAc) gave excellent agreement, with a root-mean-square-error (RMSE) of 0.3 kcal/mol, which corresponds to an average error in dissociation constants of less than two-fold (Supplementary Fig. 3). The low salt condition (0.75 mM

$MgCl_2$, 30 mM KOAc) was used to develop and test the model as the wider range of affinities that could be measured provided more extensive data to develop and test the quantitative binding model. The binding data from RNA-MaP agreed well with reliable literature affinity measurements for PUF4 and PUM2 (Supplementary Fig. 5 and refs. 34, 41).

## PUF4 and PUM2 predictive binding models differ in one flipping term

We previously developed a binding model for PUM2 containing three classes of free energy terms (Fig. 3a): (i) terms for binding at each recognition site ($\Delta\Delta G_b^X$; $b = 1–9$, representing the position of the bound residue to eight Pumilio repeats and an additional 3' site that provides a small contribution to binding; $X$ = A, C, U, or G is the base at that position); (ii) terms for binding non-contiguously, which we refer to as flipping terms ($\Delta\Delta G_f^Y$; $f = 1/2$, 2/3, 3/4, 4/5, 5/6, or 6/7 corresponds to either a single residue $Y$ = A, C, U, or G inserted between two positions or any two residues $Y$ = NN flipped between these positions); and (iii) coupling terms to account for non-additive energetics when the value of a particular energetic term depends on the residue or residues present at one or more other positions ($\Delta\Delta G_c^Z$; $c$ is the position of the coupled bases and $Z$ is the combination of residues that meet all of the conditions, see also ref. 34). For PUM2, we provided evidence for 59 needed terms, as follows: (i) 36 $\Delta\Delta G_b^X$ terms ($36 = 4 \times 9$); (ii) 20 flipping terms ($\Delta\Delta G_f^Y$), corresponding to flips of each of the four bases inserted at four different positions in the linear bound sequence ($f$ = 3/4, 4/5, 5/6, and 6/7) and NN ($20 = 4 \times 4 + 4 \times 1$); and (iii) three coupling terms ($c$ = 5–8, 6–8, or 8–9; $\Delta\Delta G_c^Z$)[34]. While this is a large number of variables, their values were constrained by precise binding data for 5206 RNAs—nearly 100 measurements per variable on average, as well as by the absence of measurable binding for ~10,000 additional RNAs. More terms are possible but are not needed to provide an excellent fit to this large dataset. For example, there are no terms in the PUM2 model for flips between positions 1 and 2 or between 2 and 3, because these flips are so energetically unfavorable that we no longer see measurable binding for these RNAs. (Our data thus define minimum free energy penalties for flips at these sites, and terms can be added as discrete values if weaker affinities can be measured in the future). The 7/8 flipping term is absent for a different reason; since all residues have similar binding at position 8, it costs less energetically to accommodate whatever residue is present in the sequence than to pay the cost to flip a residue. Finally, only three coupling terms were required, and each was modest in its value, in keeping with intuition from structural inspection of binding of RNA residues to each of the eight distinct Pumilio repeats (Fig. 1b). A strength of quantitative models like that in Fig. 3a rests in their ability to be tested, refined, and extended as additional binding data become available, in the form of minor adjustments to the current parameters or new higher-order (coupling) terms. In contrast, qualitative models are accepted or rejected often without clear criteria and without a path to refine the model or knowledge of how to redefine the model.

We applied the predictive PUM2 binding model to the 6180 PUF4 binding measurements obtained herein to determine how similar vs. distinct the binding landscape of PUF4 is relative to PUM2 (Fig. 3b). When the PUM2 model was fit to PUM1 binding data, it was shown to quantitatively account for the binding landscape for PUM1 with affinities for 3674 RNAs predicted within experimental error[34]. When we applied the PUM2 model to the PUF4 binding data, the RMSE was considerably higher than that between replicates (1.43 kcal/mol versus ~0.3 kcal/mol; Fig. 3 and Supplementary Fig. 3) and a large fraction of the data had high z-scores (2298 of 6180 constructs (37%) with z-score >3). Indeed, the near-zero $R^2$ value from linear regression indicates a poor fit of the model with no better predictive value than the null hypothesis of a horizontal line. The striking asymmetry of the error, with many more sequences binding stronger than predicted,

suggested that, of the altered or new terms, at least one term would need to be more favorable for PUF4 than PUM2.

As the PUF4 motifs are longer than PUM2 yet both proteins contain the same number of Pumilio repeats (Fig. 1a, b), a reasonable model to account for the differences would be if energetics for the individual sites ($\Delta\Delta G_b^X$ terms) remained the same, as they do for PUM1, and one or more of the flipping terms ($\Delta\Delta G_f^Y$) changed. Fitting the model with variable flipping free energies to the PUF4 data gave a substantial improvement, with the RMSE reduced by approximately two-fold, an $R^2$ value of 0.71 indicative of strongly predictive model, and far fewer points with high z-scores (Fig. 3c). Remarkably, only one term differed substantially from the PUM2 model. Whereas residue flips were strongly disfavored at all positions for PUM2, flipping was strongly favorable at position 6/7 for RNA bound to PUF4 but not at any of the other positions (Supplementary Table 1 and Table 1 below). We discuss this finding and its implications in more detail after further consideration of the data fitting.

Fitting the PUF4 binding data to the PUM2 model with all parameters varied improved the fit (Fig. 3d versus 3c), with the RMSE decreasing from 0.71 to 0.55 kcal/mol and with fewer outliers (554 versus 246 with z-score >3, respectively), and both models provided large improvements relative to the unaltered PUM2 model, which gave RMSE of 1.43 kcal/mol with 2298 outliers with z-scores of >3 (Fig. 3b). While there was some variation in the best-fit values for the individual site PUM2 and PUF4 terms, these values strongly correlated (Fig. 4a, b, blue points, $R^2$ = 0.86 and see below); the only clear difference was the ~0.9 kcal/mol discrimination against A at position 5 for PUF4 but not for PUM2 (Fig. 4a and Table 1; see also Fig. 5 below). The flipping terms were also similar except for the terms for the single residue 6/7 flip, which strongly deviated and were favorable for PUF4 but unfavorable for PUM2 as noted above (Fig. 4c, d, closed & open red points). These results support and refine the above findings of highly similar binding by PUF4 and PUM2 except for non-contiguous binding effects at position 6/7.

## Further test of the PUF4 binding model

The agreement of the model of Fig. 3c with the PUF4 data might lead one to conclude that this model is correct. However, the complexity of

### Table 1 | Thermodynamic parameter values for the PUF4 additive non-consecutive model

| Term I | $\Delta\Delta G_b^X$ (kcal/mol) $X$ = | | | |
|---|---|---|---|---|
| Bound residue position $b$ = | A | C | U | G |
| 1 | 3.69 | 2.85 | 0.00 | 3.97 |
| 2 | 2.50 | 4.41 | 4.39 | 0.00 |
| 3 | 2.67 | 3.01 | 0.00 | 2.52 |
| 4 | 0.00 | 2.29 | 1.75 | 1.51 |
| 5 | 0.89 | 0.03 | 0.00 | 1.10 |
| 6 | 0.00 | 0.93 | 0.85 | 1.41 |
| 7 | 2.48 | 1.82 | 0.00 | 2.07 |
| 8 | 0.00 | 1.47 | 1.33 | 1.33 |
| 9 | 0.60 | 0.37 | 0.00 | 0.24 |

| Term II | $\Delta\Delta G_f^Y$ (kcal/mol) $Y$ = | | | | |
|---|---|---|---|---|---|
| Flipped residue position $f$ = | A | C | U | G | NN[a] |
| 3/4 | 1.63 | 1.51 | 1.60 | 1.21 | >1.50[b] |
| 4/5 | >2.00 | 1.36 | 1.75 | >2.00 | >1.50 |
| 5/6 | >2.00 | 1.83 | 1.68 | >2.00 | >1.50 |
| 6/7 | −1.15 | −1.07 | −1.34 | −0.35 | 0.67 |

[a]2-nt flip of any sequence.
[b]>indicates a lower limit.

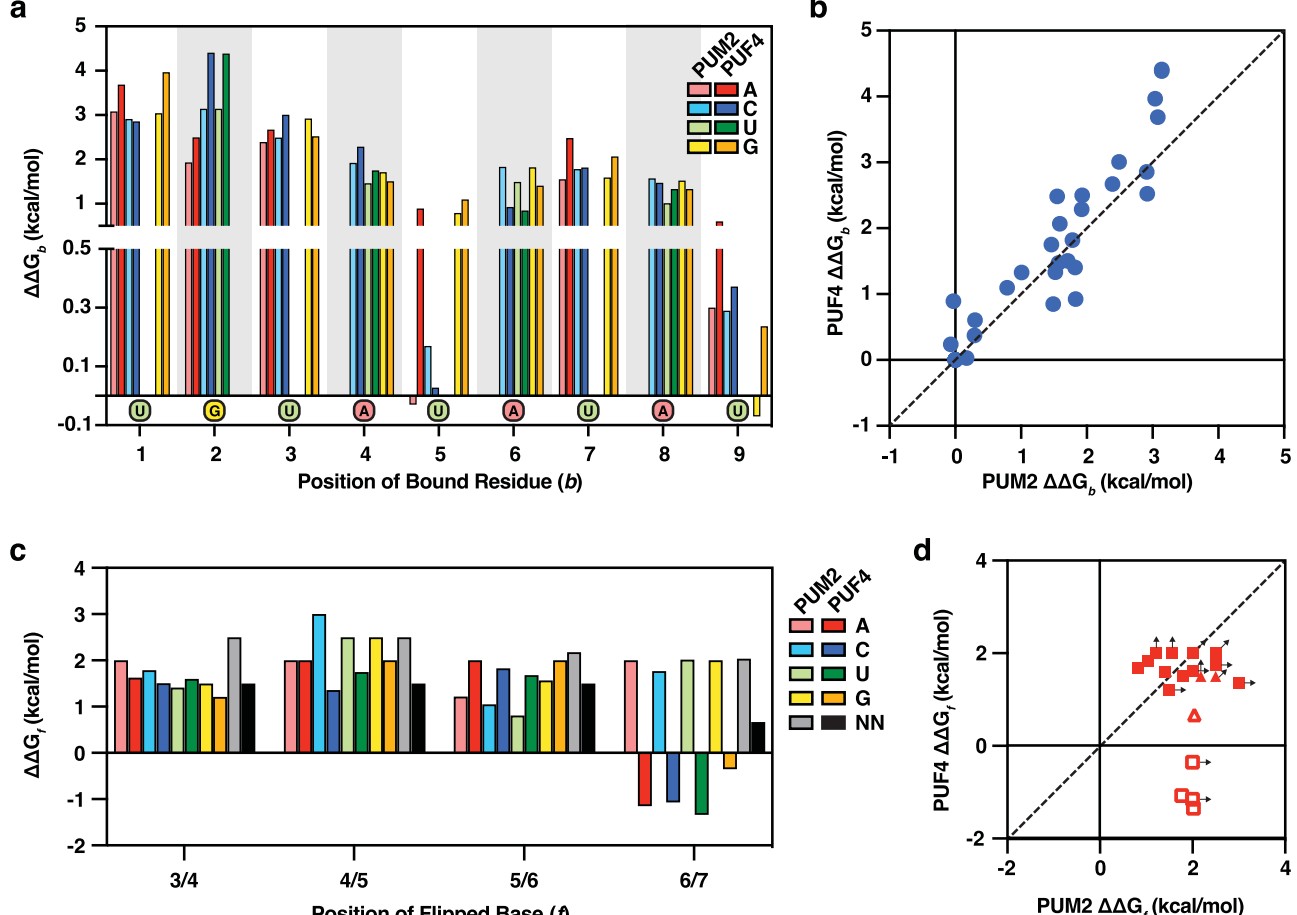

**Fig. 4 | Analysis of individual parameters for PUM2 and PUF4 predictive thermodynamic models. a** PUF4 and PUM2 additive model parameters, optimized for each RBP (from Table 1). **b** Scatterplot of all bound terms ($\Delta\Delta G_b^X$) from the PUM2 and PUF4 predictive models. The dashed line corresponds to x = y ($R^2 = 0.86$; RMSE = 0.5 kcal/mol). **c** Flipping parameters ($\Delta\Delta G_f^Y$) for PUF4 and PUM2 optimized with the data for each. NN indicates any two bases flipped at the given position (see also Table 1). **d** Scatterplot of base flip terms ($\Delta\Delta G_f^Y$) in the PUM2 and PUF4 predictive models. Open symbols are for flips at position 6/7 (single flips denoted by squares and double flips denoted by triangles) and closed are for flips at the 3/4, 4/5, or 5/6 positions (see Fig. 3a for nomenclature). Limits are shown with arrows in the figure and values are from Table 1 and ref. 34.

large datasets and the models used to fit them call for additional scrutiny. Indeed, whenever variable terms are added, fits will improve and one can, at least some of the time, get excellent fits from incorrect models. Misfitting can occur even with far more data than the number of model variables, because the data at hand typically will not equally probe each parameter. Our designed libraries provide some insurance against this limitation, relative to random or natural variants; nevertheless, one cannot design a library that equally tests all model parameters without already knowing the values for those parameters. For example, the inclusion of the coupling terms in the PUM2 model was necessary to account for a handful of clear outliers but their inclusion had a negligible effect on the overall RMSE because the terms apply to a miniscule fraction of the total RNAs and provide modest adjustments even for those[34]. Indeed, while there is likely some degree of local coupling for PUF4 as well, we did not include coupling terms in the PUF4 models because there was too little systematic data to confidently add such terms. Nevertheless, the absence of these terms may account for the somewhat higher RMSE for the final PUF4 model relative to replicate measurements and relative to the prior PUM2 model (Fig. 3d; Supplementary Fig. 3b[34]; 0.55, 0.30, and 0.34 kcal/mol, respectively).

To directly address these concerns, we carried out several additional tests of our PUF4 model. We calculated the sensitivity of the fit to each parameter (Figs. 5, 6), which showed that the values obtained

were well defined. We then further demonstrated the similarity of the individual site free energy terms ($\Delta\Delta G_b^X$) for PUF4 and PUM2 by comparing their single mutant penalties (Fig. 4 and Supplementary Fig. 6a, b).

Sensitivity analysis was carried out by varying the $\Delta\Delta G$ value of each of the 56 terms in the PUF4 model individually and determining the effect of that variation on the overall RMSE of the fit. Figures 5, 6 compares the best fit value (the minima of the curve) and sensitivity of the fit (i.e., change in RMSE) to variation in each parameter of the PUF4 and PUM2 models. Focusing first on the individual site terms ($\Delta\Delta G_b^X$, Fig. 5), the residue preferred at each site was the same in all cases for PUF4 and PUM2 (Fig. 5, red box), and the penalties for variation from the preferred residue were also highly similar. For both proteins, there were generally larger deleterious effects for residues at the 5′ region of the binding site. (The physical origins of these specificity differences are not known[34]). As expected, the sensitivity was asymmetric, especially for large deleterious substitutions where their effects often lead to difficult-to-measure or unmeasurable binding such that the minimum value of the penalty is better defined. In addition, because these substitutions are rarely found in the RNAs with measurable binding, they have a smaller impact on the overall RMSE. Nevertheless, reasonable estimates are obtained in most instances as binding is observed for some RNAs with each substitution (see also Supplementary Fig. 6). The flip terms ($\Delta\Delta G_f^Y$) were also similar for PUF4 and

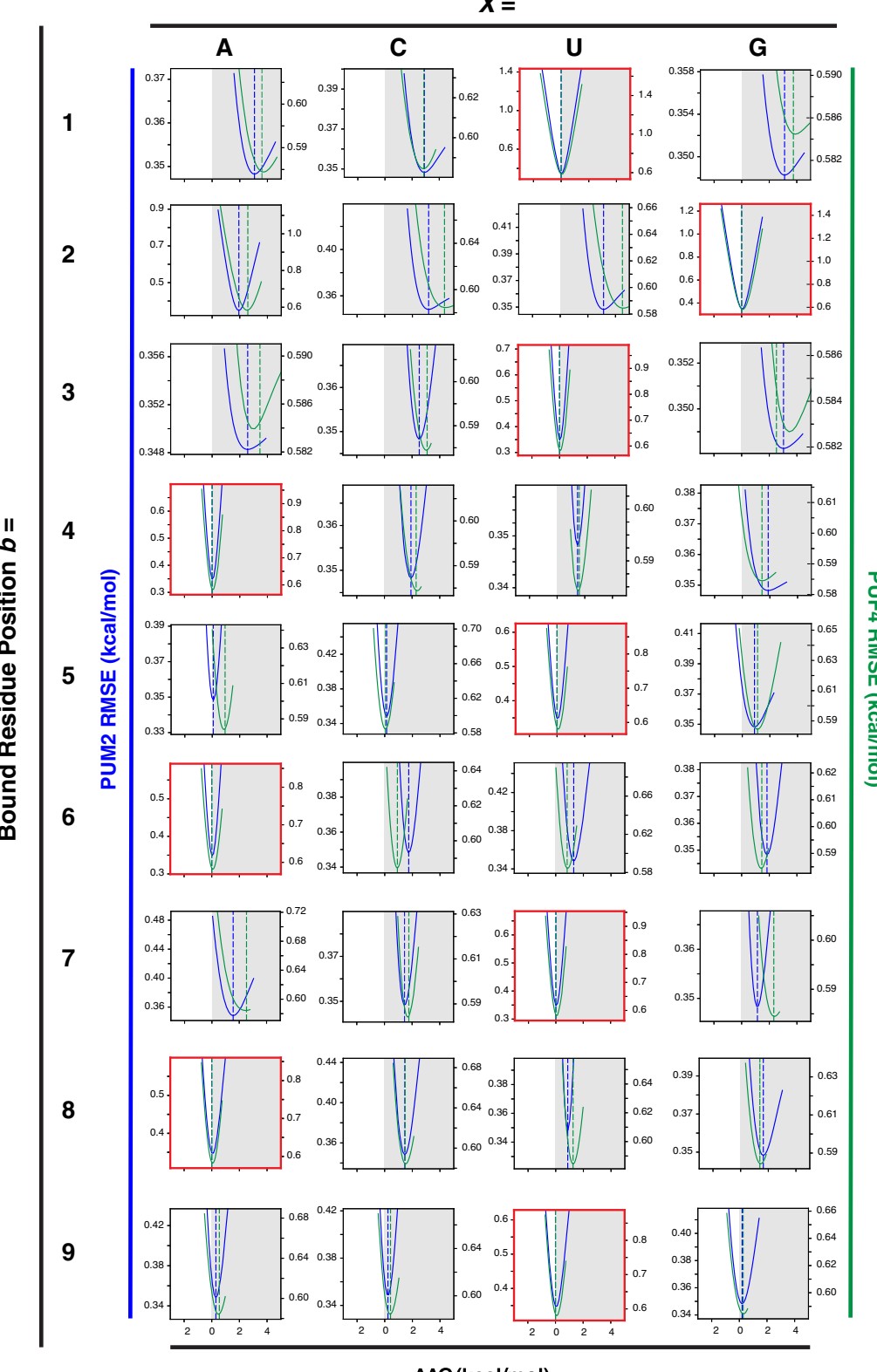

**Fig. 5 | Dependence of the overall fit (RMSE) on the individual fit bound parameters in the PUF4 and PUM2 models.** The parameter sensitivity of each bound term for the individual sites ($\Delta\Delta G_b^x$; see Fig. 3a) was determined by varying the $\Delta\Delta G$ value (x axis) of one parameter while keeping the other parameters constant (see "Methods" for details). PUM2 parameter sensitivities (blue curves, left axis) were replotted from ref. 34 and compared to parameters for the PUF4 model (green curves, right axis) following the same fitting procedure. Vertical dashed blue and green lines indicate the best fit values for each model (PUM2 and PUF4, respectively) and were similar to the minima obtained in the sensitivity analysis. Red boxes indicate the consensus or reference sequence at each position.

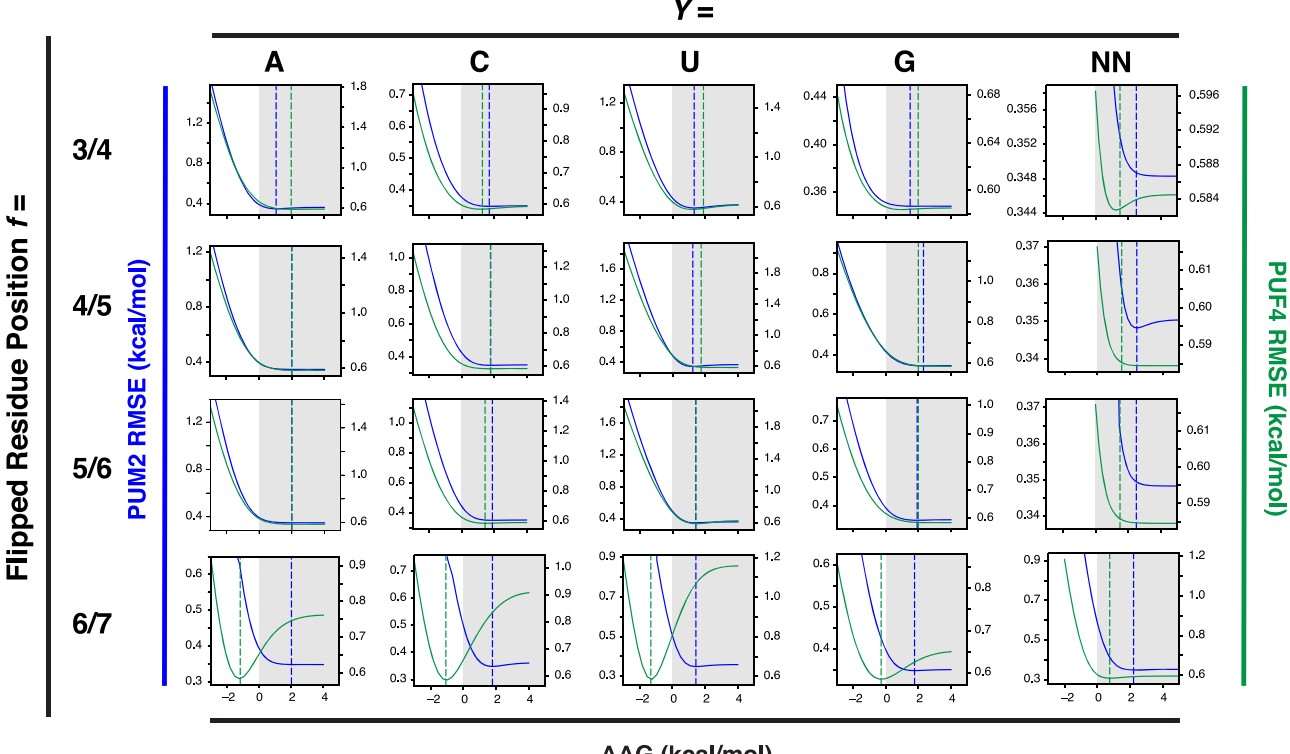

**Fig. 6 | Dependence of the overall fit (RMSE) on the individual fit flipping parameters in the PUF4 and PUM2 models.** Parameter sensitivity for flipping terms ($\Delta\Delta G_f^y$; see Fig. 3a) for the PUF4 model were calculated as in Fig. 5 (green curves, right axis). PUM2 parameter sensitivities were replotted from ref. 34 (blue curves, left axis). Vertical dashed blue and green lines indicate the best fit values for each model (PUM2 and PUF4, respectively) and were similar to the minima obtained in the sensitivity analysis.

PUM2, with the marked exception of the flips at the 6/7 position as noted above (Fig. 6). Flipping at this position is clearly favorable (negative ΔΔG) for PUF4 and clearly unfavorable (positive ΔΔG) for PUM2. Similarly, flipping two residues at the 6/7 position for PUF4 is less unfavorable than for PUM2 for two-residue flips (Table 1). The sensitivity analysis supports the conclusions above of highly similar binding rules (energetics) for PUM2 and PUF4, with the marked exception of the single favorable flip for PUF4.

To further dissect the individual site free energy terms ($\Delta\Delta G_b^x$), we compared single mutations for PUF4 and PUM2 relative to their respective consensus sequences. Each single mutation was made in four different scaffolds to control for structural effects, although not all scaffold variants were present on the analyzed chip due to bottlenecking in library generation (see "Methods"; ref. 34). This comparison again showed similar effects for PUF4 and PUM2, with average observed differences of 0.44 ± 0.4 kcal/mol (Supplementary Fig. 6b). The library also contained variants in the two residue 5′ and 3′ of the consensus sequence; these gave no 5′ preferences, as for PUM2, and modest preferences for the first position 3′ of the consensus site that were also similar to what was observed for PUM2 (Supplementary Fig. 6d and ref. 34). Thus, a more direct look at a subset of the data provides additional support for the derived PUF4 binding model and its similarity to that for PUM2. These data also support similar energetics for the individual PUM2 and PUF4 sites, consistent with their structural and sequence similarity (Fig. 1b, c).

## Discussion

We have gone beyond motif descriptions for RNA recognition and provided a mathematical model that is quantitative and predictive for RNA occupancy by the RNA binding protein (RBP) PUF4 from *S. cerevisiae*. Comparison of this model to that previously derived for PUM2[34]

provides insights into RNA recognition by these RBPs and identifies the origin of their specificity differences.

The motif descriptions for PUF4 RNA recognition are more complex than those for PUM2 (Fig. 1a). However, the mathematical descriptions for their binding thermodynamics are of similar complexity. The extensive differences that are observed in their binding specificities arise essentially from a difference in a single term that accounts for flipping out a residue between two Pumilio recognition sites (6 and 7; Fig. 3a); this term is favorable for PUF4 but unfavorable for PUM2 (Fig. 4c and Table 1). The benefit of the thermodynamic and mathematical approach is further underscored by its ability to account for the change in specificity of an engineered PUM1 variant, where a change in a (different) single model parameter provides a quantitative description for binding by the engineered variant despite binding differences of >1 kcal/mol for one in five RNA sequences[34].

Binding motifs have been obtained for many RBPs via traditional genomic experiments or variations thereof[42–44]. But many are highly degenerate and thus have limited predictive power, and machine learning approaches to date have also yielded motifs with low information content. The superiority of mathematical modeling based on accurate thermodynamic data over motif descriptions is apparent at two levels. First, a motif description is, by definition, a linear, energetically additive model−i.e., binding of residues in a set order with site energetic contributions that are independent of the other residues that are present. But even the Pumilio proteins that appear to have modular recognition elements require terms for non-linear binding ($\Delta\Delta G_f^y$ terms, Fig. 3a) to accurately account for their specificities (ref. 34 and this work). Although one can expand motif models by defining multiple motifs, this approach leaves unclear how to weight the relative affinities across them. Second, while our Pumilio binding models require little-to-no coupling to be highly predictive,

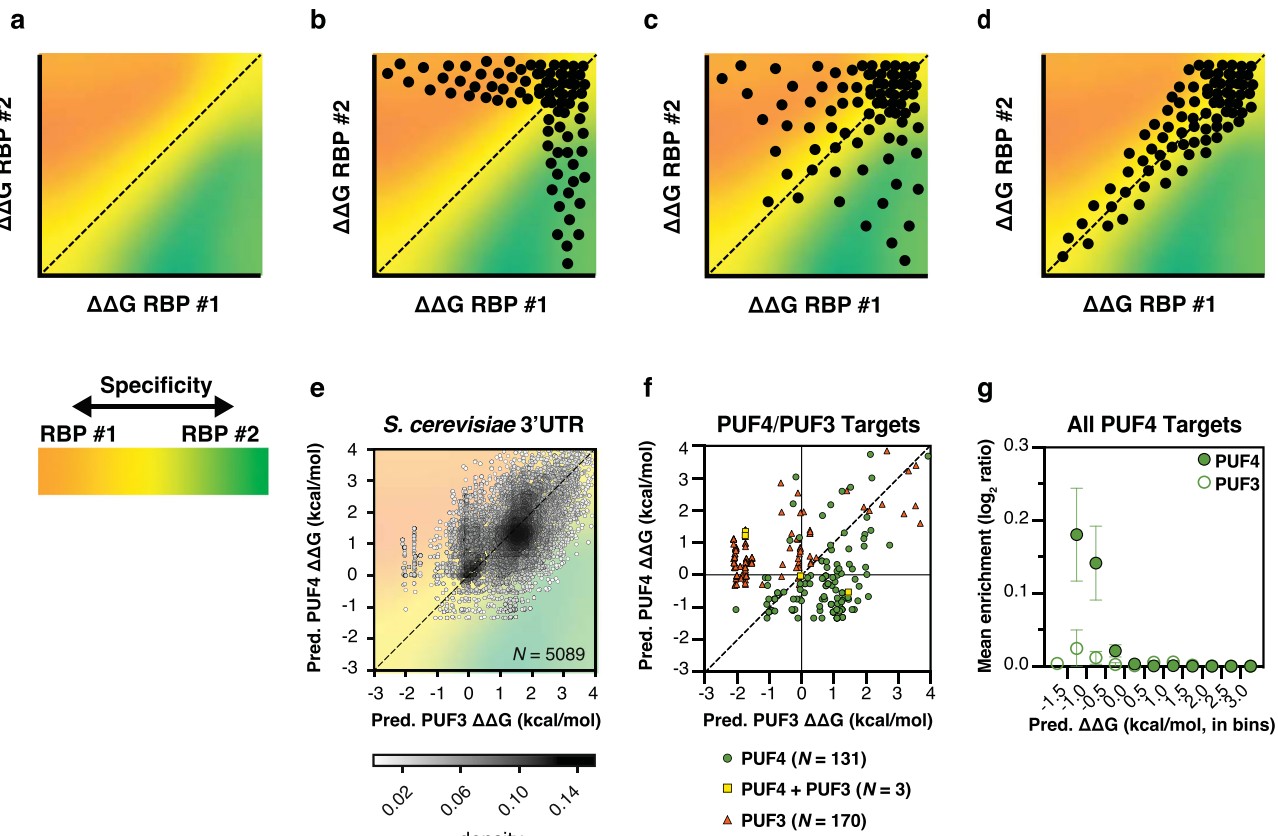

**Fig. 7 | Affinity–specificity landscapes for RNA binding proteins (RBPs). a** The landscape of RNA affinity for two RBPs, where the color coding indicates the specificity (bar under plot). **b**–**d** Mock comparison of affinities for two RBPs binding, RBP #1 (orange) and RBP #2 (green), to all RNAs in a transcriptome. Three plots are shown that represent a continuum of possibilities, ranging from fully distinct targets (**b**) to identical specificities (**d**) with **c** showing an intermediate case with specificity overlap. Plots in **a**–**d** are in terms of ΔΔG, which provides affinities relative to a reference RNA. **e**, **f** PUF4/PUF3 affinity–specificity landscapes for the highest affinity binding site for the respective RBP of all *S. cerevisiae* 3′UTRs in the Saccharomyces genome database (**e**)[56] and for PUF3 and PUF4 3′UTR targets identified from previous in vivo experiments (**f**)[9]. The mathematical binding models from this and prior work[34] were used to calculate the highest affinity binding site within the 3′UTR of each transcript. To obtain estimates for PUF3 using the highly homologous PUM2 data, an additional term was added to account for PUF3's

favorable interaction with C at position −2 (see Supplementary Table 2 and "Methods")[51]. Values are shown as ΔΔG relative to the respective consensus sequences. Points are given for each binding site (N = 5089) and shaded based on the density of points with the corresponding value. In **f**, PUF3 targets are depicted by an orange triangle; targets that bind PUF3 and PUF4 with a yellow square, and PUF4 targets with a green circle. N is the number of known targets for each. **g** Thermodynamic affinity predictions compared to PUF4 RNA-IP enrichment from *Saccharomyces cerevisiae*[9]. The enrichment (log₂ ratio) for PUF4 targets was reported in ref. 9 for significantly enriched targets (from four independent experiments). The enrichment was estimated as 0 for sites in the yeast genome with predicted affinities <4 kcal/mol that were not identified in the target list. The mean of the enrichment was taken for bins of 0.5 kcal/mol with error bars indicate 95% CIs of the mean.

mathematical models can be expanded, as demanded by current or future data, to include coupling terms and ensure predictive power. RBPs with less regular recognition elements than the Pumilio proteins and with less defined motifs (Fig. 1a, b) will likely require more coupling terms (Fig. 3a, $\Delta\Delta G_c^Z$), reflecting RNA-RNA interactions in bound complexes and interconnected binding modes.

As implied above, the form of the mathematical model needed to describe RNA binding is rooted in the fundamental physical properties of these interactions. The free energy terms for each base binding to each site in PUM1/2 and PUF4 are nearly the same, consistent with the high degree of conservation in their binding interfaces (Fig. 1, Supplementary Fig. 1)[22,45]. Nevertheless, the sequence changes in each Pumilio repeat could have altered specificity, and we would not have known that they do not without explicit, quantitative models to compare. In addition, within the same Pumilio protein, repeats with the same recognition amino acids that favor interactions with the same RNA residue, nevertheless do so with different degrees of specificity (Supplementary Fig. 1 and ref. 34). In this way, mathematical models reveal thermodynamic properties that require biophysical studies to uncover molecular explanations.

Our mathematical model showed that recognition specificity is the same or nearly the same within each of the Pumilio proteins. Instead, the specificity arises from the energetic terms for binding continuously (PUM1/2) versus non-continuously (PUF4) from repeat 6 to repeat 7 (Figs. 1b and 4c, d). These observations are consistent with prior motif assignments that showed that PUF4 tends to bind longer RNA motifs than PUM2 (Fig. 1a)[9,10,26,27,36].

Structural studies of PUF4 and PUF5, which recognize longer RNAs than PUM2 and PUF3, suggest a lessened curvature of the Pumilio crescent shape (Fig. 1b), and it has been suggested that the lessened curvature is responsible for the preference for binding RNAs with internal flips and thus longer sequences[39,45,46]. Structures of PUF4 bound to 9 nt RNA targets also support non-contiguous binding, where a C base is flipped between positions 5 and 6[47] or a U base is flipped between 6 and 7[45]. Interestingly, our results indicate that 5/6 flipping is disfavored and 6/7 flipping is favored (Table 1). This difference underscores that an X-ray crystallographic structure captures a snapshot of a distribution and that the particular RNA sequence used, conditions, and local packing interactions can affect the binding mode that is observed. Our determination of a specific position of favorable

flipping for PUF4 and flipping penalties that are similar to those for PUM2 at other sites suggests that altered curvature, if responsible for the specificity difference, manifests its thermodynamic effects at a particular site. Additional large-scale thermodynamic specificity studies with additional Pumilio protein variants and mutants will be required to test models for their specificity, specificity differences across similar Pumilio repeats, and how flipping of residues to yield non-contiguous binding is favored and discouraged at different positions and in different Pumilio proteins.

We next turn to broader biological implications and introduce a specificity–evolvability paradox. High specificity in biology is typically considered to be a favorable trait, in line with how one would engineer a high-precision system; the more specific an RBP, the better defined its targets and the more precise its regulation can be. However, increased specificity lessens the likelihood that new RBP functions will evolve, as random mutations in RNA sequence are less likely to significantly increase RBP occupancy at the newly mutated sites. In this scenario, the selective advantage needed to guide the evolution of a protein for a new regulatory role may require multiple deleterious RNA mutations and thus be of low probability. Thus, there is a tension between specificity and evolvability, where evolvability describes the ability to effectively traverse a fitness landscape to find alternative fitness peaks[48].

To further explore this paradox, consider the binding specificities of two hypothetical RBPs that we describe by the affinity–specificity landscape in Fig. 7a. In one extreme (Fig. 7b), the RBPs have highly distinct specificities, so that the strong binders to RBP1 bind weakly to RBP2 (points in orange region) and vice versa (points in green region); there is no significant overlap in binding (yellow) until binding is negligible for both. In this case, the transition of RNA targets between the RBPs is unlikely, as multiple simultaneous changes in the target sequence are likely needed to provide a selective advantage. However, the Pumilio proteins PUM2/PUF3 maintain a common binding specificity throughout evolution, but their target sequences have changed, with at least five events in eukaryotic evolution leading to distinct RNA target sets[28]. The observed near-continuum of binding affinities across sequences close in Hamming space has likely been key in allowing these transitions[28,34]. In the other extreme (Fig. 7d) the affinities fall along the diagonal and there is no specificity—all RNAs bind RBP1 and RBP2 with the same affinity; in this case, there is regulation by both RBPs rather than by RBP1 or RBP2. This scenario appears to hold with human PUM1 and PUM2[34] and likely other Pumilio proteins[49,50]; presumably multiple RBPs with shared specificity allows same or similar sets of RNAs to be regulated at different times or locations, or in different ways.

In reality, there is a continuous distribution of possible RBP specificities, with the scenario shown in Fig. 7c representing an intermediate between those in Fig. 7b, d, and this distribution might be expected to be representative of most RBPs in nature. Our mathematical models allowed us to explore this conjecture with affinity–specificity landscapes for natural RBPs, and the affinity–specificity landscape for PUF3/PUF4 fits the intermediate regime (Fig. 7e). (We use PUF3, a close homolog of PUM1/2, because it is present along with PUF4 in *S. cerevisiae* and appears to have a highly similar binding landscape)[31,51]. The highest affinity sites prefer either PUF3 or PUF4, as is important for biological specificity and consistent with their largely distinct target sets[9]. Nevertheless, there are numerous 3′UTRs that share similar high affinities for PUF3 and PUF4, implying that there is the potential to share binding and potentially use that overlap to shift specificities in the face of a selective pressure. Indeed, a remarkable switch occurred in a branch of fungal evolution, where >100 PUF3 targets transitioned to control by PUF4[28,52]. The change in regulation from one Pumilio RBP to another is predicted to be more probable than a change in regulation by an unrelated RBP because of their overlapping specificity (Fig. 7e). Indeed, the highest

affinity PUF3 and PUF4 sites within a given 3′UTR are sometimes the same and sometimes different (see also Supplementary Fig. 7f, g), and the high density of sites of similar and intermediate affinity (centered on the diagonal at around −1.5 kcal/mol in Fig. 7e) arises because of the high probability of finding a UGUA sequence (the common Pumilio strong binding motif) within a 3′UTR.

Figure 7f shows the affinities and specificities for the PUF3 and PUF4 RNA targets[9] identified from in vivo experiments, calculated using our mathematical model. On average, the enrichment of PUF4 targets follows the thermodynamic affinity predictions of this model (Fig. 7g), analogous to what was reported for PUM2[34]. These results also meet our general expectation that the strongest PUF3 (orange) and PUF4 (green) binders tend to be targets of each (Fig. 7f). However, there is a set of PUF3 targets with similar predicted affinities for PUF3 and PUF4 (orange; Pred. ΔΔG ~0 kcal/mol), and a small number of the PUF3 and PUF4 targets are predicted to bind weakly (upper right quadrant). Also, of the three targets that appear to be common, one has similar strong affinity, but the others are predicted to prefer PUF3 or PUF4 (yellow squares). These outliers may represent limitations in quantitating occupancy from genomic pull-down and cross-linking approaches and/or cellular features that alter binding specificities. These observations underscore the power of comprehensive thermodynamic measurements, as carried out herein, to provide clear predictions and the future challenge to develop approaches to accurately and quantitatively report RNA/RBP occupancies in cells.

## Methods

### Library design
An RNA library was designed as outlined in Fig. 2a and Supplementary Fig. 2a. Complete sequence information of each variant is provided in the Source data.

### Library preparation and sequencing
Methods for amplification, assembly, and sequencing of the library are reported in ref. 34. Sequences for primers and oligos referenced below can be found in ref. 34. See also Supplementary Fig. 2b.

In brief, oligonucleotides were first amplified using emulsion PCR (ePCR) to give a diverse library pool (64–130 nt length). Libraries were then fractionated by size on an 8% polyacrylamide gel into six fractions (based on UV visualization of marker lanes). Purified fractions were re-amplified using the Read2 and RNAPstall_adapt primers. Final constructs for each library fraction were assembled by PCR using C_read1_bc_RNAP, D_read2, OligoC, and OligoD primers, as illustrated in Supplementary Fig. 2a. The C_read1_bc_RNAP primer incorporates a unique molecular identifier (UMI), a randomized 15 nucleotide barcode, to check for sequencing errors[34]. The library was then diluted by bottlenecking to ~700,000 total molecules. The final library was sequenced using MiSeq Reagent Kit v3 on Illumina MiSeq instrument.

### Protein expression and purification
The RNA-binding domain of *S. cerevisiae* PUF4 (537–888) was first cloned into a pET28a-based expression vector (New England Biolabs) between a His-tag and TEV cleavage site at the N terminus and a SNAP tag at the C-terminus. For protein expression, constructs were transformed into *E. coli* BL21 (DE3) strain (Agilent) and induced with 1 mM IPTG, between an OD600 of 0.6–0.8 and at 22 °C for 18–20 h. Cells were pelleted by centrifuging at 5000 × $g$ at 4 °C for 15 min and resuspended in lysis buffer A (20 mM Na-HEPES (Spectrum), pH 7.4, 500 mM KOAc (Spectrum), 5% glycerol (ThermoScientific), 0.2% Tween-20 (Sigma), 10 mM imidazole (Sigma-Aldrich), 2 mM dithiothreitol (DTT, Sigma-Aldrich), 1 mM phenylmethylsulfonyl fluoride (PMSF, ThermoScientific), and 1X Complete Mini protease inhibitor cocktail (Roche)). To ensure efficient lysis, cells were passed through four Emulsiflex (Avestin) cycles. The mixture was centrifuged at 20,000 × $g$, 4 °C for 20 min, separating the supernatant from the cell

debris. Polyethylene imine (Sigma) was added dropwise to the supernatant to a final concentration of 0.21% (v/v) over 30 min at 4 °C and constant agitation to precipitate nucleic acids. The mixture was centrifuged at 20,000 × g and 4 °C for 20 min. The resulting supernatant was filtered through a 0.25 μm syringe filter. FPLC (Bio-Rad) was used to purify the PUF4 construct from the cleared lysate. The solution was loaded onto a Nickel-chelating HisTrap HP column (GE) and washed using Buffer A excluding protease inhibitors. The protein was eluted over a 10–500 mM imidazole gradient over 1 h using Buffer A and B (20 mM Na-HEPES (Spectrum), pH 7.4, 500 mM KOAc (Spectrum), 5% glycerol (ThermoScientific), 0.2% Tween-20 (Sigma), 500 mM imidazole (Sigma-Aldrich)). Protein fractions were pooled and desalted using a HiPrep 26/10 desalting column (GE) into Buffer C (20 mM Na-HEPES, pH 7.4, 50 mM KOAc, 5% glycerol, 0.1% Tween-20, 2 mM DTT). The His-tag was removed with 0.63 mg TEV protease (NEB) by incubating for 13 h at 4 °C while agitating. The His-tag was removed by a second HisTrap HP column purification, eluting over a 0–500 mM imidazole gradient using Buffer C and B to separate protein from cleaved adapter. The purified protein fractions were again pooled and desalted into Buffer C using a HiPrep 26/10 desalting column (GE). The purified protein was next loaded onto a HiQ column and eluted over a gradient of 50 mM–1 M KOAc using Buffer C and D (20 mM Na-HEPES, pH 7.4, 1 M KOAc, 5% glycerol, 0.1% Tween-20, 2 mM DTT). Protein fractions were pooled and desalted into Buffer E (20 mM Na-HEPES, pH 7.4, 100 mM KOAc, 5% glycerol, 0.1% Tween-20 and 2 mM DTT) and concentrated using Amicon Ultra-0.5 filters (Milipore). The PUF4 solution was diluted two-fold with Buffer E with 80% glycerol (final concentration 43%) and stored at −20 °C until use.

### Cy3B-labeling of SNAP-tagged proteins

PUF4 was labeled with Cy3B-BG (product of coupling Cy3B-NHS, GE, and BG-NH2, NEB, as described in ref. [34]) for RNA-MaP binding experiments following a similar protocol to that described by ref. [34].

SNAP-tagged PUF4 was labelled with Cy3B in Buffer C by combining 20 μM Cy3B-BG and 10 μM of purified SNAP-PUF4 stock at 4 °C for 12–14 h in the dark while rotating. Free dye was removed with a 7 kDa Zeba Spin Desalting Columns (Thermo Fisher Scientific). The fluorescently-labelled protein was concentrated using an Amicon Ultra-0.5 filter (10 kDa, Milipore). For storage at −20 °C, the solution was diluted two-fold with Buffer C with 80% glycerol as described above.

### RNA-MaP equilibrium binding experiments

Equilibrium binding experiments were performed as previously described[34]. In total, three equilibrium binding experiments were performed with PUF4 protein. Two measurements were taken on chip 1, testing the effects of salt concentration on binding and an additional replicate experiment of high salt on chip 2. We use the high salt to define replicate error, which is similar to that previously found for PUM2[34]. We derive the binding model using the low salt data as that provides the largest range of affinities and thus the most information for developing our model. Data from all experiments are available in Source data.

Two salt concentrations were tested varying the MgCl₂ and KOAc concentrations in the binding buffer: low (0.75 mM MgCl₂, 30 mM KOAc) and high (2 mM MgCl₂, 100 mM KOAc). All other reagents remained constant in each buffer (20 mM Na-HEPES, pH 7.4, 0.1% Tween-20, 5% glycerol, 0.1 mg/mL BSA, and 2 mM DTT). The library on the chip was sequentially equilibrated with increasing concentrations of Cy3B-labelled PUF4. Two-fold serial dilutions were prepared with PUF4 in binding buffer at concentrations ranging from 0.0125 to 409.6 nM in addition to a no-protein control. Dilutions were stored at 4 °C in the dark until use. Each solution was pumped into the flow cell sequentially and incubated at 25 °C from 33 min at the lowest protein

concentration to 19 min for the highest protein concentration, taking fluorescent measurements after each addition.

### Determining active protein fraction by titration

Cy3B-labeled, SNAP-tagged PUF4 (10–100 nM final) was incubated with a saturating concentration of unlabeled consensus RNA (20 nM; (AUGUGUAUAUUAGU; Integrated DNA Technologies (IDT), Coralville, IA)) and trace ³²P-labeled RNA of the same sequence (<0.1 nM) for 35 min (25 °C) under high salt conditions (2 mM MgCl₂, 100 mM KOAc, 20 mM Na-HEPES, pH 7.4, 0.2% Tween-20, 5% glycerol, 0.1 mg/mL BSA and 2 mM DTT). After equilibration, 7.5 μL aliquots were transferred to 5 μL of ice-cold loading buffer, containing 6.25% Ficoll PM 400 (Sigma), 0.075% BPB, and 2.5 μM unlabeled consensus RNA in binding buffer. The samples were loaded on a pre-chilled 20% native acrylamide gel and run at 42 V/cm constant voltage at 5 °C, in 0.5x Tris/Borate/EDTA (TBE) running buffer (44.5 mM Tris-borate, 1 mM Na₂EDTA, pH 8.3). The gel was dried, exposed to a phosphorimager screen, and scanned with a Typhoon 9400 Imager. The binding signal was quantified using TotalLab Quant (TotalLab, Newcastle-Upon-Tyne, UK), and fitting was performed with KaleidaGraph 4.1 (Synergy Software, Reading, PA; RRID:SCR_014980). The active protein fraction (69%) was determined based on the intersection of lines fit through protein concentrations above and below the breakpoint.

### Sequencing data processing

Sequencing data was processed as previously described in ref. [34]. Briefly, the tile identifier and x-and y-locations for each cluster were computationally derived from the fastq sequencing output as outlined in refs. [40,53]. Sequencing clusters were divided into three categories (1) RNA library of interest, (2) fiduciary marker, and (3) PhiX background DNA based on alignment of the read1 sequence to (1) RNA polymerase initiation site and stall sequence TTTATGCT ATAATTATTTCATGTAGTAAGGAGGTTGTATGGAAGACGTTCCTGGAT CC, (2) fiduciary marker sequence CTTGGGTCCACAGGACACTCG TTGCTTTCC, or (3) neither, respectively. Only RNA-encoding clusters were used for $K_D$ fitting. Library variats were assigned to sequencing data clusters using a unique molecular identifier (UMI, barcode in Supplementary Fig. 2b) as described in ref. [34].

### Fluorescence normalization and determination of the free energy of binding

The protein-bound fluorescence (green channel) was normalized to the amount of RNA transcribed (red channel) for a given cluster. The normalized fluorescence values at varied protein concentration were used to estimate the equilibrium dissociation constant $K_D$ through a multi-step fitting procedure to allow robust fitting over a range of affinities as previously described in ref. [34]. First, the normalized fluorescence values were fit to a binding curve to obtain best-fit values for $K_D$ for each variant and variant-independent parameters $f_{min}$, $f_{max}$, and $K_{D,NS}$. Next, the distribution of variant-independent values was used to refine these parameters for all variants. Finally, the variant-independent variables were used to refine an estimate for $K_D$ for each variant.

The binding model includes a term for non-specific binding where a second protein monomer can bind to the RNA–protein complex at high protein concentrations, as was observed previously for PUM2 (ref. [34], Eq. 1).

$$\text{R} + \text{P} \underset{}{\overset{K_D}{\rightleftharpoons}} \text{R} \cdot \text{P} \underset{}{\overset{K_{D,NS}}{+ P \rightleftharpoons}} \text{R} \cdot \text{P} \cdot \text{P} \tag{1}$$

where $R$ is RNA, $P$ is protein, $K_D$ is the dissociation constant ($K_D = e^{\Delta G/RT}$), and $K_{D,NS}$ is the non-specific dissociation constant ($K_{D,NS} = e^{\Delta G_{NS}/RT}$).

The normalized fluorescence is related to the protein concentration [P] by the following equation:

$$f = f_{min} + f_{max} \frac{[P]}{[P] + K_D} \left( 1 + \frac{[P]}{[P] + K_{D,NS}} \right) \quad (2)$$

## Single-cluster fitting

Python's least-squares fitting package (v0.8.3) was used to fit the above binding model to the normalized fluorescence values of each individual cluster. During fitting all parameters ($K_D$, $f_{min}$, $f_{max}$, and $K_{D,NS}$) were allowed to vary to find the optimal fit. The initial estimates and constraints are as follows: $f_{min}$ was initialized as the median fluorescence values of all clusters at the lowest applied protein concentration and constrained to be greater than or equal to zero; $f_{max}$ was initialized as the maximum observed fluorescence value for each individual cluster and constrained to be greater than or equal to zero; $K_{D,NS}$ was initialized as five-fold the highest applied protein concentration; $K_D$ was initialized as the highest protein concentration.

## Defining variant-independent parameters

To increase the confidence of the $K_D$ initially fit above, the parameters $f_{min}$, $f_{max}$, and $K_{D,NS}$ were estimated based on the per-variant values for each individual dataset, where the per-variant values are the median of the single-cluster values associated with the same molecular variant. The value for $f_{min}$ was largely consistent across variants (Supplementary Fig. 2d). Thus, the median value across all variants was used as an estimate for $f_{min}$. The values for $f_{min}$ are similar for different salt concentrations, with the following values: 0.0134 for the low salt concentration and 0.0159 and 0.0388 for the high salt replicate concentrations. The value of $f_{max}$ varied widely for variants with weak affinities where protein binding was not saturating at the concentrations tested. In contrast, the $f_{max}$ value was consistent for high-affinity variants (Supplementary Fig. 2e). To find common values for $f_{max}$, a subset of variants with low and precisely measured $K_D$ values based on the single-cluster fits were used to fit a gamma distribution as outlined in ref. 34. Variants with low $K_D$ values were defined as precise if they had a standard error of $\Delta G < 1$ kcal/mol, a standard error on the fit $f_{max}$ less than $f_{max}$, and an initial $K_D$ value that was <15% of the highest protein concentration tested. The significance of the data filtering was assessed by rejecting the null hypothesis that 25% of clusters would pass all of these filters by chance alone (binomial $p$ value < 0.01). $K_{D,NS}$ values do not vary considerably for non-saturating variants, similar to $f_{min}$ (Supplementary Fig. 2f). The values for $\Delta G_{NS}$ are similar for different salt concentrations, with the following values: −8.49 kcal/mol for the low salt concentration and −8.30 and −8.58 kcal/mol for the high salt replicate concentrations.

## Determining $K_D$

A refined estimate of $K_D$ was determined by refitting a single binding curve to the median fluorescent cluster values at each protein concentration per variant, using the fixed variant-independent parameters for $f_{min}$ and $K_{D,NS}$ from above. For variants that did not achieve near-saturation, defined as the median fluorescence values at the highest concentration of protein being less than the 99.7% confidence interval of the fit $f_{max}$ distribution, $f_{max}$ was resampled from the variant-independent distribution. For variants that achieved saturation, $f_{max}$ was allowed to vary. To obtain error estimates for $K_D$, we resampled clusters of a given variant 100 times, fitting the median values with each iteration to determine a 95% confidence interval. The confidence interval was used to determine an error:

$$\Delta G_{error} = \Delta G_{upper\ bound} - \Delta G_{lower\ bound} \quad (3)$$

## Data filtering

The $K_D$ values were used to calculate the free energy of binding, $\Delta G$, where $\Delta G = RT\ln(K_D)$ (Source data). Variants were included in our analysis if they met the following criteria: (1) Variants with $\Delta G_{error} < 1.0$ kcal/mol, (2) Five or more clusters per variant in each experiment and replicate, and (3) observed $\Delta G$ values less than −7.69 kcal/mol, where more than 15% of RNA was bound at the highest protein concentration.

## Combining replicates

A small shift (0.1 kcal/mol) in the mean difference between the highest affinity binders was observed between replicate datasets. This was corrected by subtracting the offset from the second replicate dataset. Duplicates were then combined as described by computing the error-weighted mean:

$$\Delta G_{comb} = \left( \frac{\Delta G_1}{\sigma_1^2} + \frac{\Delta G_2}{\sigma_2^2} \right) \left( \frac{1}{\sigma_1^2} + \frac{1}{\sigma_2^2} \right)^{-1} \quad (4)$$

and the combined error computed via error propagation:

$$\sigma_{comb} = \sqrt{ \left( \frac{1}{\sigma_1^2} + \frac{1}{\sigma_2^2} \right)^{-1} } \quad (5)$$

## Filtering out structured variants

PUF protein binding is affected by RNA structure[34,41]. In the initial derivation of the PUM2 thermodynamic model, sequences with predicted structure were filtered out. For this we used Vienna RNAfold (v. 2.4.14) command "RNAfold -p0 -T25 -C" and constrained the full length of the designed RNA variants to be single stranded with the exception of the stem when using Scaffold S2a and S2b, which was truncated, e.g.:

UUCUUUCUUGUAUAUUAUUUCUUUCU
XXXXXXXXXXXXXXXXXXXXXXXXXX

which was compared with the fully unconstrained sequences equally truncated:

UUCUUUCUUGUAUAUUAUUUCUUUCU
..........................

Variants were then removed based on their structural stability:

$$\Delta G_{fold} = \Delta G_{unconstrained} - \Delta G_{constrained} \quad (6)$$

with a cutoff of $\Delta G_{fold} < -0.5$ kcal/mol, giving 6180 high precision affinity variants.

## Determining relative affinities

Affinities were determined relative to the linearly bound UGUAUAUAU sequence. For the reference sequence, an additional structural cutoff of $\Delta G_{fold} < -0.2$ kcal/mol was used before choosing the median value of all occurrences across the library ($n = 237$) as reference affinity. The $\Delta\Delta G$ for all variants were calculated, where:

$$\Delta\Delta G_{obs} = \Delta G_{variant} - \Delta G_{UGUAUAUAU,median} \quad (7)$$

## Assessing alternative binding registers in single mutants

PUF4 can bind in any available register of the RNA variant. To assess the binding affinity of each alternative registers, the full sequence of

each single mutant was split into overlapping 9-mer windows shifted by a single base. Affinity of each window was determined via the equation below using an additive consecutive model, where $X$ is the residue (A, C, U, or G), $b$ is the position of the bound residue (1–9), and $\Delta\Delta G_b^X$ the penalty as ascribed by each mutation relative to the PUF4 consensus UGUAUAUUA. Alternative binding registers were determined by comparing the sequence of the window with highest affinity with the sequence of the designed register. All alternative registers and their associated affinities are listed in Supplementary Fig. 6c. Alternative binding registers were only observed for mutants at position 2, each of which had predicted affinities within 0.5 kcal/mol of experimentally observed values and are within experimental error.

$$\triangle\triangle G_{\text{pred}} = \sum_{b=1}^{9} \triangle\triangle G_b^X \tag{8}$$

### Development, testing, and evaluation of PUF4 thermodynamic binding models

PUF4 binding was predicted for all 6180 filtered sequences by calculating $\Delta\Delta G_{\text{pred}}$ using the $\Delta\Delta G_b^X$ and $\Delta\Delta G_f^Y$ penalties from the PUM2 additive non-consecutive model described in ref. 34. Due to the high stability of the hairpin stem for variants with scaffold S2a and S2b, only the sequence of the loop region was considered. Affinities are then predicted by computing the relative ensemble affinity of each individual variant.

$$\triangle\triangle G_{\text{ensemble}} = - \text{RT} \ln\left(\sum_{r=1}^{n-9} e^{\frac{-\triangle\triangle G_r}{\text{RT}}}\right) \tag{9}$$

The relative ensemble affinity is given by the partition function, summing over all possible registers and modes. Modes include: additive consecutive binding ($\sum_{r=1}^{9} e^{\frac{-\triangle\triangle G_{r,\text{consec}}}{\text{RT}}}$), single flips ($\sum_{r=1}^{10} e^{\frac{-\triangle\triangle G_{r,1-\text{nt,flip}}}{\text{RT}}}$), two consecutive flips ($\sum_{r=1}^{10} e^{\frac{-\triangle\triangle G_{2\text{flips}}}{\text{RT}}}$), and two independent flips ($\sum_{r=1}^{10} e^{\frac{-\triangle\triangle G_{2-\text{ntflip}}}{\text{RT}}}$). The predicted values (Pred. ΔΔG) were compared to the experimentally observed values (Obs. ΔΔG) in Fig. 3b.

Next, the PUM2 model was refined by allowing all flip parameters to vary and optimize by minimizing the sum of the squared error between the predicted and measured ΔΔG values. The $\Delta\Delta G_f^Y$ parameters were optimized using the lmfit package (version 0.8.3) and the BFGS optimizer, initializing each parameter at the PUM2 parameter values. Flip penalties were allowed to vary from −4 to 4 kcal/mol to find an optimal value. The final penalties for $\Delta\Delta G_f^Y$ can be found in Supplementary Table 1. The improved fit was compared to Obs. ΔΔG in Fig. 3c.

Finally, all the parameters were optimized except for the consensus bases UGUAUAUAU, which were held at 0 kcal/mol. Bound penalties ($\Delta\Delta G_b^X$) were initialized as their single mutant values for UGUAUAUUAU and excluding the values at position 7. Their optimization was restricted to the larger of the two 95% CI error bounds or +/−0.75 kcal/mol. Single mutants $\Delta\Delta G_3^A$, $\Delta\Delta G_3^G$, and $\Delta\Delta G_5^G$ could not be determined accurately because they either fell below the structure cutoff ($\Delta G_{\text{fold}} < -0.5$) or not satisfying any of the other cutoffs as laid out in section "Data filtering" (Supplementary Fig. 6b). These values were allowed to vary with a larger bound of +/−2.5 kcal/mol. Flip parameters were initialized at 0 kcal/mol and allowed to vary from +/−4 kcal/mol. To prevent overfitting, the data was split into equivalently sized and randomly drawn testing and training sets. The model was first optimized on the training set before applying the derived parameters on the testing dataset. Supplementary Fig. 4b shows equivalent fits between both the training and test set with an RMSE of 0.55 kcal/mol and $R^2$ of 0.84. The predicted values (Pred. ΔΔG) were compared to the experimentally observed values (Obs. ΔΔG) in Fig. 3d.

A parameter sensitivity analysis was performed on the parameters of the PUF4 model (Figs. 5, 6). A single parameter was allowed to vary

at a time while the other values were fixed. The value was then varied within the chosen upper and lower bound and the RMSE of Obs. ΔΔG versus Pred. ΔΔG calculated. Parameters that did not show a distinct minima were reset to the value where the initial plateau was reached.

### BLAST alignment

The DNA sequence for the yeast PUF4 (Uniprot P25339) homology domain was used as the query sequence for a protein-protein alignment with BLAST (blastp). The PUF4 sequence was aligned to the homology domains for yeast PUF3 (Uniprot Q07807) and human PUM2 (Uniprot Q8TB72).

### Predicting PUF3 and PUF4 binding sites in the yeast genome

The *S. cerevisiae* genome was obtained from the Saccharomyces Genome Database (www.sgd-archive.yeastgenome.org/sequence/S288C_reference/genome_releases/, version R62-1-1).

The forward and reverse strands were each divided into 11 nt sliding windows, shifting by a single base. PUF3 (an ortholog of PUM2) and PUF4 binding affinities were predicted for each 11 nt window using the PUM2 or PUF4 models, respectively, to determine the relative ensemble affinity, $\Delta\Delta G_{\text{ensemble}}$ to each consensus sequence (UGUAUAUAU for PUF3/PUM2 and UGUAUAUUAU for PUF4). Each ensemble affinity is the partition function of binding in every possible register consecutively or flipped, following the equation below:

$$\triangle\triangle G_{\text{ensemble}} = - \text{RT } \ln\left(\sum_{r=1}^{9} e^{\frac{-\triangle\triangle G_{r,\text{consec}}}{\text{RT}}} + \sum_{r=1}^{10} e^{\frac{-\triangle\triangle G_{r,1-\text{nt,flip}}}{\text{RT}}} + e^{\frac{-\triangle\triangle G_{2-\text{ntflip}}}{\text{RT}}} + e^{\frac{-\triangle\triangle G_{2\text{flips}}}{\text{RT}}}\right) \tag{10}$$

The highest affinity register for each 11 nt windows was sorted with all of the windows from highest affinity to lowest. For any window with an overlap in sequence to a window higher on the list, the lower window was removed. Windows with ΔΔG < 4 kcal/mol were kept for further analysis, resulting in 407,329 and 419,354 binding sites for PUF3/PUM2 and PUF4, respectively. Comparisons across models were made by matching the affinities for each model for the same window.

### Predicting PUF3 and PUF4 binding sites in the *S. cerevisiae* genome with a longer sequence motif

PUF3 has a similar binding motif to PUM2 with an additional preference for bases CN upstream of the 5′ end of the UGUAUAUAU consensus. To more accurately predict binding to PUF3, two additional positions were included in the mathematical model at position −2 and −1 to the 5′ end, yielding an additional 8 parameters, 4 single mutants at each position. Parameters were estimated based on PUF3 binding affinities calculated in ref. 51. The $K_D$ values were converted to ΔG using $\Delta G = -\text{RT} \ln K_D$ and relative affinities computed with respect to the published UGUAAAUA affinity (add references). The final single mutant parameter for C at position −2 was set to −2 kcal/mol with all other additional parameters set to 0 kcal/mol (see Supplementary Table 2).

The upstream preference for PUF3 requires an extension of the genome analysis window by two bases. The method described above was repeated with a 13 nt sliding window. To allow for a comparison to the PUF4 model, its motif was also extended by two bases upstream with all 8 single mutant parameters set to zero. After filtering, 511936 and 494532 binding affinities with ΔΔG > 4 kcal/mol were calculated for PUF3 and PUF4, respectively.

### 3′UTR annotation for yeast transcriptome

The 3′UTR annotations for *S. cerevisiae* were downloaded from the Saccharomyces Genome Database (www.sgd-archive.yeastgenome.org/sequence/S288C_reference). The annotation was analyzed in Python and chromosome, start, stop, and sequence information collated into a BED file and compared to the yeast genome version R62.

Duplicates and overlapping annotations were removed by selecting the longest annotated sequence. Binding sites for PUF3 and PUF4 were determined by counting all filtered binding windows within the annotated sequence or selecting the highest affinity binding window as required.

### Predicting the binding affinity of known PUF4 and PUF3 targets in *S. cerevisiae*

A list of PUF4 and PUF3 targets was taken from ref. 9. For target sites in the 3′UTR, the gene location was compared to predicted binding affinities from the 13 nt window analysis described above. The highest affinity (lowest ΔΔG) site was determined in each 3′UTR for PUF3 and PUF4 from the windows described above.

### Reporting summary

Further information on research design is available in the Nature Research Reporting Summary linked to this article.

## Data availability

Source data for Figs. 2c, 3b–d, including the thermodynamic binding data generated in this study are available in Source data. All other data supporting the findings of this study are available from the corresponding author on request. Source data are provided with this paper.

## Code availability

Key scripts for the analyses reported here have been deposited at https://github.com/HerschlagLab/PUF4Model.git.

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

## Acknowledgements

Financial support was provided by the National Institutes of Health (R01 GM132899 to D.H.), We thank Greg Hogan and Traci Hall for the thoughtful comments on the manuscript and the Herschlag laboratory for discussion and review of the manuscript.

## Author contributions

D.H., W.J.G., I.J., and P.P.V. conceived the project and experimental design. I.J. and P.P.V designed the RNA library, performed the RNA-MaP experiment, and purified the PUF4 protein. S.K.D. derived the fluorescent values to determine binding affinities. C.S., S.K.D., and W.R.B. wrote the code for obtaining binding affinities. C.S. performed the model fitting and parameter sensitivity analysis with help from W.R.B. L.D.H. aligned and analyzed the PUF4 and PUM2 crystal structures and sequences. C.S. applied the Pumilio models to the yeast transcriptome. L.D.H. compared the Pumilio models to the in vivo Pumilio targets. D.H., L.D.H., and C.S. wrote the manuscript with input from all authors.

## Competing interests

The authors declare no competing interests.
