## [Peer Review File · Nature Communications]

A comprehensive thermodynamic model for RNA binding by the *Saccharomyces cerevisiae* Pumilio protein PUF4Reviewers' Comments:

Reviewer #1:

Remarks to the Author:

In this manuscript the authors report a complete quantitative model to describe the RNA binding specificity of the RNA binding protein PUF4. They use HTS-MaP and sequence specificity modeling that was first applied to MS2, but then more practically applied for pumilio repeat proteins PUM1/2. The current study extends these ground breaking analyses of pumilio repeat proteins to the related PUF4 protein. The resulting sequence specificity model is well supported by the data which are high throughput analysis of ca. 6000 sequence variants.

The authors report that the specificity model for RNA binding of yeast PUF4 is highly similar to PUM1/2. The major finding of this study is that residue flipping is unfavorable for PUM1/2 known from previous work, however modeling of the affinity distribution data shows that flipping is favorable at specific positions for RNA binding by PUF4. A key strength of the modelling is its simple connection to RNA structure and physical interactions, and thus physical attributes like base flipping can be identified and better understood. Authors make the case that standard optimal sequence logos do not explain RBP specificity. Transcriptomic analysis of RBP binding sites can never provide a complete picture since the transcriptome represents only a small fraction of sequence space, and limited structural contexts for any one sequence. Predictive models require the kinds of affinity distribution data that methods like HTS-MaP provides. The determination of a binding distributions for large sequence populations for any RBP is very challenging, few have been analyzed in this detail. The continued successful application of specificity modeling for pumilio repeat proteins that includes position scores, interaction terms, and accommodates bulged or "flipped" positions is an important advance. The lessons learned investigating pumilio proteins, although their structure is uncharacteristically modular compared to other RBPs, are nonetheless highly valuable and instructive for other researchers investigating nucleic acid binding specificity. This is exceptionally timely and interesting research that uses cutting edge experimental and conceptual methods. The subject matter of RBP specificity is incredibly important especially given the overall importance of RNA biology as well as the challenges inherent to describing RBP specificity.

There are two issues that I think the authors should consider, and then revise their manuscript accordingly.

1. The level of support for the conclusions is excellent. An area of concern however are validation experiments measuring the binding affinity with individual RNA ligands. Supplementary Fig. 5 shows only three examples where HTS-MaP results and modeling are compared to binding affinity assays of individual RNA sequences. For the two weakest binding sequences the results correspond. But HTS-MaP and individual binding results are way off for the one tight binding sequence (10-100 fold different? I must be missing something). Also, the variants chosen do not appear to represent logically chosen examples of flipped out or non-flipped out sequences, which would validate the key conclusions of the high throughput assay and specificity modeling. Given the appropriately rigorous approach to binding assays espoused by several of the authors (ref47, a classic) this aspect of the analysis seems notably less well developed.

2. P6 last paragraph. From the text, methods, and supplementary information the comparison it can be appreciated that the comparison of technical replicates was done at a different salt concentration than the experiment used to develop the model. This may be well justified, but I think the justification and details here must be stated clearly, the rationale behind using difference salt concentrations more clearly explained in the main text for the reader. A concise outline of what is known about the influence of salt concentration for PUF4 and potential pitfalls in interpretation given that different salt concentrations were used. Also, if the effects of salt concentration on binding were compared between experiments were new insights gained?

Minor-

1. The numerous panels in Fig. 5 are described only briefly in the text. Can they be moved to supplemental information? Perhaps a more detailed description to guide the reader toward visualizing what is important in the figure would be helpful.
2. Supplementary Fig. 3. First line – Puf4 should be PUF4
3. Fig. 3 Strongly suggest putting the “obs” and “pred” in much larger font in the figures where these terms are used as subscripts.

Reviewer #2:

Remarks to the Author:

The manuscript by Sadee & Hagler et al. describes the deep profiling of binding affinities for the RNA binding protein PUF4 in order to develop a comprehensive binding model for how PUF4 interacts with RNA, and how this binding is distinct from family members PUM2 and PUM1. I think this method and the altered binding model is quite interesting, and the analyses of the RNA library and subsequent binding models build well off this prior work and show the power of continued use of this approach to deeply map RNA binding protein binding affinities.

My only major criticism is that the manuscript is largely limited to analysis of the RNA-MaP dataset, and has limited validation of the findings with orthogonal approaches. For me, one of the most powerful analyses in the prior Jarmoskaite et al. PUM1/2 paper was the comparison of RNA-MaP-based binding affinity with an orthogonal binding assay (eCLIP data), which confirmed that the developed binding model for PUM2 was not specific to this in vitro system but rather was also recapitulated in vivo. However, as far as I can tell, the only such data in this manuscript is Sup. Fig. 5, which is limited to 3 specific 9-mers (with the canonical motif having a ~100-fold variation between the three publications).

I would find this manuscript far stronger if there was additional explicit testing and orthogonal validation of the key findings (particularly the binding difference to flip vs non-flip 6/7 sequences between PUM2 and PUF4), either biochemically or from CLIP or other data. The Gerber et al. 2004 PLoS Biology publication included RIP-microarray profiling, which is obviously somewhat lower resolution than CLIP, but perhaps it would be sufficient to see this distinction? Otherwise the confidence in the results of the paper rests heavily on one's confidence that this method is sufficiently validated by the Jarmoskaite paper to trust the results of its application to a different (albeit highly related family member) protein.

Minor comments:

While Fig 6. a-d would be appropriate for the discussion section, to me Fig. 6e-f (and the associated text) are more 'results' than 'discussion' and I would suggest separating the presentation of results (the analysis of 3'UTR sequences) from the broader discussion topics.

Although I appreciate that the authors tried to condense much of the model visualization in Fig. 3a, for me this figure became a bit too crowded – in particular, I found it much easier to understand what a 'flipped' interaction looked like once I referred back to the model in the original Jarmoskaite et al. PUM1/2 paper. As the position 6/7 flip difference is one of the key findings in this paper, I think it is especially important that this paper have clarity on what that interaction looks like.

Perhaps I'm confused what exact datasets are being used, but it seems like the UGUUAUUA has a Kd of 1.37 nM in Fig. 2c, but then Sup. Fig. 5 has it as ~0.8 or so. Is this a difference in showing one

scaffold (Fig. 2c) vs an average of all (Sup. Fig. 5)? Either way I think it should be clarified what exactly is being shown in Sup. Fig. 5.

Reviewer #3:

Remarks to the Author:

Herschlag and co-workers report a quantitative thermodynamic model for RNA binding by the yeast protein PUF4. The authors use a previously developed high throughput approach, RNA-MaP, to simultaneously measure equilibrium binding of the yeast protein PUF4 to 15,272 distinct RNAs. They obtain reliable affinities for 6,180 RNAs and use these data to devise a thermodynamic model that explains these affinities. A notable feature of this model are specific terms for "flipped" bases, which explains recognition of non-contiguous RNA bases. These "flipping" terms of PUF4 differ from corresponding "flipping" terms for the related PUM2 protein, for which the authors previously devised a similar model.

The presented work is carefully executed. The data are of exceptional quality, well described and expertly placed in a fundamental, evolutionary context (i.e., RNA binding site "evolvability"). The study is an important contribution to our understanding of RNA biology and therefore of interest to a wide readership.

Comments:

1) The RNA-MaP affinities correspond to a small select number of other PUF4 affinity measurements (Suppl.Fig.5), but not to other published data. The authors provide a reasonable explanation for their selection. However, it is not clear whether data from ref. 47 refer to other RNA-MaP data or to measurements with an orthogonal approach, which is an important control as the RNA-MaP measurements are performed with a fluorescently labeled protein. In this respect, the discrepancy to one of the two values of ref 61 is notable, and raises potential questions about distortions introduced by the fluorescent label. Many high throughput studies benchmark their data with "traditional" biophysical approaches, but those experiments are not included here. The authors might want to clarify and perhaps test whether or not the measurements could be affected by the fluorescent label inherent to the approach.

2) Affinities for 6,180 RNAs, while an impressive number, represent only a bit more than 2% of the entire sequence space of the 9-mer PUF4 binding site (> 262k sequence variants). The obtained model clearly describes the obtained affinities well and is carefully checked against overfitting (Suppl. Fig.4). However, since the data only cover a small sliver of the high affinity end of the binding spectrum, it might be premature to call this a "comprehensive" model. The vast majority of the sequencing space of the 9-mer binding site is not included in the model, if I understand the model correctly. This terminology needs to be clarified. The described model is excellent for the high affinity part of the PUF4 affinity spectrum, but perhaps not "comprehensive".

We thank the reviewers for the comments and suggestions. We were pleased that the reviewers shared our perspective on the urgent need for comprehensive thermodynamic models for RNA-protein interactions. We have made changes to the manuscript to address each of the reviewers' concerns, as elaborated below, including a deeper discussion of the validation of the RNA-MaP method and comparison of the thermodynamic predictive model to an in vivo dataset. Overall, we believe that these edits clarify aspects of the manuscript and will make it more readable and accessible.

Reviewer comments in black

Responses in blue

Changes to the text in red

Reviewer #1 (Remarks to the Author):

In this manuscript the authors report a complete quantitative model to describe the RNA binding specificity of the RNA binding protein PUF4. They use HTS-MaP and sequence specificity modeling that was first applied to MS2, but then more practically applied for pumilio repeat proteins PUM1/2. The current study extends these ground breaking analyses of pumilio repeat proteins to the related PUF4 protein. The resulting sequence specificity model is well supported by the data which are high throughput analysis of ca. 6000 sequence variants.

The authors report that the specificity model for RNA binding of yeast PUF4 is highly similar to PUM1/2. The major finding of this study is that residue flipping is unfavorable for PUM1/2 known from previous work, however modeling of the affinity distribution data shows that flipping is favorable at specific positions for RNA binding by PUF4. A key strength of the modelling is its simple connection to RNA structure and physical interactions, and thus physical attributes like base flipping can be identified and better understood. Authors make the case that standard optimal sequence logos do not explain RBP specificity. Transcriptomic analysis of RBP binding sites can never provide a complete picture since the transcriptome represents only a small fraction of sequence space, and limited structural contexts for any one sequence. Predictive models require the kinds of affinity distribution data that methods like HTS-MaP provides. The determination of a binding distributions for large sequence populations for any RBP is very challenging, few have been analyzed in this detail. The continued successful application of specificity modeling for pumilio repeat proteins that includes position scores, interaction terms, and accommodates bulged or "flipped" positions is an important advance. The lessons learned investigating pumilio proteins, although their structure is uncharacteristically modular compared to other RBPs, are nonetheless highly valuable and instructive for other researchers investigating nucleic acid binding specificity. This is exceptionally timely and interesting research that uses cutting edge experimental and conceptual methods. The subject matter of RBP specificity is incredibly important especially given the overall importance of RNA biology as well as the challenges inherent to describing RBP specificity.

We thank the reviewer for placing the work into the perspective of the field and sharing their perspective on the need for and importance of this work.

There are two issues that I think the authors should consider, and then revise their manuscript accordingly.

1. The level of support for the conclusions is excellent. An area of concern however are validation experiments measuring the binding affinity with individual RNA ligands. Supplementary Fig. 5 shows only three examples where HTS-MaP results and modeling are compared to binding affinity assays of individual RNA sequences. For the two weakest binding sequences the results correspond. But HTS-MaP and individual binding results are way off for the one tight binding sequence (10-100 fold different? I must be missing something). Also, the variants chosen do not appear to represent logically chosen examples of flipped out or non-flipped out sequences, which would validate the key conclusions of the high throughput assay and specificity modeling. Given the appropriately rigorous approach to binding assays espoused by several of the authors (ref47, a classic) this aspect of the analysis seems notably less well developed.

We had tried to convey that a significant amount of the literature K_D values are not reliable –and indeed expected to be incorrect based on the criteria from reference 47 that the reviewer refers to –and we tried to do so without ‘calling out’ these results from other authors in the field. We apparently erred in not making this point clear enough (and have rectified this –see figure & red text below). There are two critical considerations for accurately obtaining K_D values: (1) the equilibration time and (2) the concentration of RNA. Of the 11 literature values for PUF4 binding constants, 6 (Table R1, rows 1-6) were measured with an inadequate equilibration time (as described in reference 47) and 8 of 11 measurements do not report varying the RNA concentration. Additionally, 7 of 11 measurements were obtained under different temperature conditions to those reported in this manuscript with RNA-MaP. Thus, only two reliable measurements have been obtained by traditional methods (Supplemental Fig. 5). These values span an affinity range of more than three orders of magnitude and the agreement of each with the RNA-MaP measurement is close for each (new Supplemental Figure 5 below).

There is a large amount of data from several systems that all show good agreement between RNA-MaP and traditional measurements. Specifically, on- and off-chip comparisons have been made for five other proteins with a total of 49 comparisons in addition to the two reported herein, and all show good agreement (Jarmoskaite 2019, Buenrostro 2014, She 2017, Tome 2014, Nutiu 2011). Based on this extensive proof-of-principle, we chose not to take the considerable time it would take to make additional PUF4 measurements. We also note that the RNA-MaP experiment is *more* direct and better controlled than most traditional biochemical methods (e.g., EMSA): binding is followed over time (to ensure equilibration) and directly (so that there are no distortions from subsequent dissociation, etc.), and the RNA-MaP experiment involves more replicates (at least 5 and often many more) than traditionally obtained so that statistical analysis is far more robust.

Index	Protein	RNA	K_D (nM)	Varied [RNA]?	T_{equil}	t_{equil}	Sufficient t_{equil} ?
1 ^a	PUF4	UGUAUA U UA	13.6 ± 0.9	nr	4 °C	1 h	no
2 ^a	PUF4	UGUAUA C UA	15.7 ± 4.6	nr	4 °C	1 h	no
3 ^a	PUF4	UGUAUA A UA	22.3 ± 4.6	nr	4 °C	1 h	no
4 ^a	PUF4	UGUAUA G UA	30.0 ± 7.6	nr	4 °C	1 h	no
5 ^a	PUF4	UGUAUA U UA	26.4 ± 7.1	nr	4 °C	1 h	no
6 ^a	PUF4	UGUAUAUA	31.6 ± 4.3	nr	4 °C	1 h	no
7 ^b	PUF4	UGUAUA U UA	10 ± 5	nr	25 °C	30 min	yes
8 ^b	PUF4	UGUAU G UAU	250 ± 50	nr	25 °C	30 min	yes
9 ^c	PUF4	UGUAUA U UA	0.00139 ± 0.00009	yes	0 °C	0.5-24 h	yes
10 ^c	PUF4	UGUAUA U UA	0.120 ± 0.03	yes	25 °C	0.5-4.5 h 10-110 min	yes
11 ^c	PUF4	C GUAUA U UA	204 ± 10	yes	25 °C	min	yes

Table R1. PUF4 binding affinities reported in the literature measured by traditional methods. The PUF4 consensus sequence is UGUAUAUUA. Underlined bases indicate a position where the base is flipped out. Red bases indicate a mutation from the consensus sequence. The time needed for equilibration (t_{equil}) can be determined by the k_{off} at a given temperature or by varying t_{equil} .^b nr = not reported

References: ^aMiller 2008; ^bHook 2007; ^cJarmoskaite 2020.

Supplementary Fig. 5. Comparison of RNA-MaP PUF4 affinities to literature values. Published literature affinities for PUF4 binding to RNAs measured at 25 °C, following the criteria for reliable measuring binding affinities outlined in ref. 2 (i.e., sufficient equilibration time and [RNA] varied

and $\langle K_D \rangle$), compared to the binding of similar RNA variants from this work. Other literature values for PUF4^{3,4} are not directly comparable because they were measured at different temperatures, with equilibration times not long enough to ensure equilibration, or without varying the [RNA] (see ref. 2). On- and off-chip comparisons have been made for five other proteins with a total of 49 comparisons in addition to the two reported herein, and all show good agreement.^{1,5-8}

2. P6 last paragraph. From the text, methods, and supplementary information the comparison it can be appreciated that the comparison of technical replicates was done at a different salt concentration than the experiment used to develop the model. This may be well justified, but I think the justification and details here must be stated clearly, the rationale behind using difference salt concentrations more clearly explained in the main text for the reader. A concise outline of what is known about the influence of salt concentration for PUF4 and potential pitfalls in interpretation given that different salt concentrations were used. Also, if the effects of salt concentration on binding were compared between experiments were new insights gained?

We clarify that each titration experiment has >5 technical replicates for each sequence analyzed. We performed a fully independent replicate on another chip to independently test our results and calculate the error between experiments. Reproducibility is high from our independent replicates and from prior RNA-MaP studies (Jarmoskaite 2020; Denny 2018; Jain 2017). After establishing this, we turned to a lower salt condition so that we could measure binding over a larger dynamic range and thus obtain a more complete and more extensively tested model. A future study on salt effects would be of interest and should include wider ranges of salt concentrations and types. In the absence of this more complete salt data, we have refrained from interpreting the data currently available (but will share with the reviewers that the data are consistent with each individual term being unchanged with salt, with the exception of the overall penalty term, as expected for greater screening at higher salt).

To the main text:

Equilibrium binding was measured for two salt conditions (“low” and “high;” see Methods). Fully independent datasets measured on two chips with “high” salt (2 mM MgCl₂, 100 mM KOAc) gave excellent agreement, with a root-mean-square-error (RMSE) of 0.3 kcal/mol, which corresponds to an average error in dissociation constants of less than two-fold (Supplementary Fig. 3). The “low” salt condition (0.75 mM MgCl₂, 30 mM KOAc) was used to develop and test the model as the wider range of affinities that could be measured provided more extensive data to develop and test the quantitative binding model.

To Supplementary Fig. 3:

Two technical replicates of PUF4 binding measurements under the “high” salt condition (2 mM MgCl₂, 100 mM KOAc) were fit as described in the Methods.

Minor-

1. The numerous panels in Fig. 5 are described only briefly in the text. Can they be moved to supplemental information? Perhaps a more detailed description to guide the reader toward visualizing what is important in the figure would be helpful.

We would like to maintain figure 5 and have added text that better describes this figure. While we can be brief in referring to it, figure 5 represents an important type of error analysis that is often neglected; we therefore believe there is value in highlighting this in the main text. Figure 5 also provides the clearest visual representation of the expected precision of the model terms, and we want to be sure to be transparent in how precisely each term is constrained by the data.

Sensitivity analysis was carried out by varying the $\Delta\Delta G$ value of each of the 56 terms in the PUF4 model individually and determining the effect of that variation on the overall RMSE of the fit. Fig. 5 shows the best fit value (the minima of the curve) and sensitivity of the fit (i.e., change in RMSE) to variation in each parameter of the PUF4 and PUM2 models.

2. Supplementary Fig. 3. First line – Puf4 should be PUF4

The correction has been made.

Two technical replicates of PUF4 binding measurements under the “high” salt condition (2 mM $MgCl_2$, 100 mM KOAc) were fit as described in the Methods.

3. Fig. 3 Strongly suggest putting the “obs” and “pred” in much larger font in the figures where these terms are used as subscripts.

We have moved “Obs.” and “Pred.” to before $\Delta\Delta G$ on the axes of Fig. 3 to make this clearer, as shown below.

a

b

c

d

Reviewer #2 (Remarks to the Author):

The manuscript by Sadee & Hagler et al. describes the deep profiling of binding affinities for the RNA binding protein PUF4 in order to develop a comprehensive binding model for how PUF4 interacts with RNA, and how this binding is distinct from family members PUM2 and PUM1. I think this method and the altered binding model is quite interesting, and the analyses of the RNA library and subsequent binding models build well off this prior work and show the power of continued use of this approach to deeply map RNA binding protein binding affinities.

We are glad that the reviewer appreciated the power of this approach and the value of this work.

My only major criticism is that the manuscript is largely limited to analysis of the RNA-MaP dataset, and has limited validation of the findings with orthogonal approaches. For me, one of the most powerful analyses in the prior Jarmoskaite et al. PUM1/2 paper was the comparison of RNA-MaP-based binding affinity with an orthogonal binding assay (eCLIP data), which confirmed that the developed binding model for PUM2 was not specific to this in vitro system but rather was also recapitulated in vivo. However, as far as I can tell, the only such data in this manuscript is Sup. Fig. 5, which is limited to 3 specific 9-mers (with the canonical motif having a ~100-fold variation between the three publications).

We agree with the reviewer that comparison to in vivo data is of value and extends the conclusions from this work. For this reason, we added a panel to figure 6 comparing the predicted affinities with an in vivo dataset, as described in the response to comments to Reviewer #3.

I would find this manuscript far stronger if there was additional explicit testing and orthogonal validation of the key findings (particularly the binding difference to flip vs non-flip 6/7 sequences between PUM2 and PUF4), either biochemically or from CLIP or other data. The Gerber et al. 2004 PLoS Biology publication included RIP-microarray profiling, which is obviously somewhat lower resolution than CLIP, but perhaps it would be sufficient to see this distinction? Otherwise the confidence in the results of the paper rests heavily on one's confidence that this method is sufficiently validated by the Jarmoskaite paper to trust the results of its application to a different (albeit highly related family member) protein.

We described the strong validation of RNA-MaP thermodynamic measurements in response to Reviewer #1 above, and we have clarified this in the text as well. In addition, as Reviewer #2 notes, the agreement with in vivo binding data from Gerber et al. provides further support and has been added to the manuscript in response to this comment and the comment above. The comparison strongly supports the RNA-MaP thermodynamic model and the validity of this model under cellular conditions. The specific revisions are reproduced below.

New text in **bold & underlined**:

...The change in regulation from one Pumilio RBP to another is predicted to be more probable than a change to regulation by an unrelated RBP because of their overlapping specificity (Fig. 6e). Fig. 6f shows the affinities and specificities for the PUF3 and PUF4 RNA targets⁹ identified from *in vivo* experiments, calculated using our mathematical model. **On average, the enrichment of PUF4 targets follows the thermodynamic affinity predictions of this model (Fig. 6g), analogous to what was reported for PUM2.²⁰** These results also meet our general expectation that the strongest PUF3 (orange) and PUF4 (green) binders tend to be targets of each (Fig 6f). However, there is a set of PUF3 targets with similar predicted affinities for PUF3 and PUF4 affinities (orange; $\Delta\Delta G_{\text{pred}} \sim 0$ kcal/mol), and a small number of the PUF3 and PUF4 targets are predicted to bind weakly (upper right quadrant). Also, of the three targets that appear to be common, one has similar strong affinity, but the others are predicted to prefer PUF3 or PUF4 (yellow squares). These outliers may represent limitations in determining occupancy from genomic pull-down and cross-linking approaches and/or cellular features that alter binding specificities. These observations underscore the power of comprehensive thermodynamic measurements, as carried out herein, to provide clear predictions and the future challenge to develop approaches to accurately and quantitatively report RNA/RBP occupancies in cells.

g Thermodynamic affinity predictions compared to PUF4 RNA-IP enrichment from *Saccharomyces cerevisiae*⁹. The enrichment (\log_2 ratio) for PUF4 targets was reported in ref. 9 for significantly enriched targets. The enrichment was estimated as 0 for sites in the yeast genome with predicted affinities less than 4 kcal/mol that were not identified in the target list. The mean of the enrichment was taken for bins of 0.5 kcal/mol with error bars indicate 95% CIs of the mean.

Minor comments:

While Fig 6. a-d would be appropriate for the discussion section, to me Fig. 6e-f (and the associated text) are more 'results' than 'discussion' and I would suggest separating the presentation of results (the analysis of 3'UTR sequences) from the broader discussion topics.

We reconsidered this ordering but chose to keep the original order to aid the flow of the story; in particular, the implications this analysis has for future studies.

Although I appreciate that the authors tried to condense much of the model visualization in Fig. 3a, for me this figure became a bit too crowded – in particular, I found it much easier to understand what a 'flipped' interaction looked like once I referred back to the model in the original Jarmoskaite et al. PUM1/2 paper. As the position 6/7 flip difference is one of the key findings in this paper, I think it is especially important that this paper have clarity on what that interaction looks like.

We agree and have revised figure 3 to better convey the physical model. (See revised figure above under Reviewer #1 comments.)

Perhaps I'm confused what exact datasets are being used, but it seems like the UGUUAUUA has a K_d of 1.37 nM in Fig. 2c, but then Sup. Fig. 5 has it as ~ 0.8 or so. Is this a difference in showing one scaffold (Fig. 2c) vs an average of all (Sup. Fig. 5)? Either way I think it should be clarified what exactly is being shown in Sup. Fig. 5.

There are 185 sequences in our dataset containing the UGUUAUUA consensus with different flanking sequences and embedded into different scaffolds (only considering binders within our measurable binding cutoff). We have shown that the base just downstream of the UGUUAUUA motif has an effect, although modest, on binding (see Supplemental Figure 6d, position +1). For Figure 2c a single binding curve was chosen, in this instance UGUUAUUAC, where the last base has a detrimental effect on binding ~ 0.4 kcal/mol. We updated the figure to show the binding curve for UGUUAUUUAU which is more representative of the core dissociation constant of 0.60 nM and have made the following changes to Fig. 2:

c Representative binding curves from RNA-MaP of two RNA variants (UGUUAUUUAU in S1a scaffold and UGUUAUCGCAC in S1b scaffold). Black circles indicate median fluorescence at each protein concentration for all clusters of the respective variants normalized to the background fluorescence of the RNA channel. The number of replicate clusters is denoted by n. Error bars represent 95% confidence intervals (CI) across the clusters determined by bootstrap analysis. The green lines indicate the fits to the binding model. The grey area indicates the 95% CI of the fit ($K_D(\text{consensus}) = 0.88$ nM, $CI_{95\%} = (7.86$ nM; 1.03 nM) ; $K_D(\text{mutant}) > 2.4$ μ M, corresponding to the upper limit for binding affinities that could be confidently distinguished from background).

Reviewer #3 (Remarks to the Author):

Herschlag and co-workers report a quantitative thermodynamic model for RNA binding by the yeast protein PUF4. The authors use a previously developed high throughput approach, RNA-MaP, to simultaneously measure equilibrium binding of the yeast protein PUF4 to 15,272 distinct RNAs. They obtain reliable affinities for 6,180 RNAs and use these data to devise a thermodynamic model that explains these affinities. A notable feature of this model are specific terms for “flipped” bases, which explains recognition of non-contiguous RNA bases. These “flipping” terms of PUF4 differ from corresponding “flipping” terms for the related PUM2 protein, for which the authors previously devised a similar model.

The presented work is carefully executed. The data are of exceptional quality, well described and expertly placed in a fundamental, evolutionary context (i.e., RNA binding site “evolvability”). The study is an important contribution to our understanding of RNA biology and therefore of interest to a wide readership.

We thank the reviewer for their positive comments, and we are glad that the importance of this work was clearly conveyed.

Comments:

1) The RNA-MaP affinities correspond to a small select number of other PUF4 affinity measurements (Suppl.Fig.5), but not to other published data. The authors provide a reasonable explanation for their selection. However, it is not clear whether data from ref. 47 refer to other RNA-MaP data or to measurements with an orthogonal approach, which is an important control as the RNA-MaP measurements are performed with a fluorescently labeled protein. In this respect, the discrepancy to one of the two values of ref 61 is notable, and raises potential questions about distortions introduced by the fluorescent label. Many high throughput studies benchmark their data with “traditional” biophysical approaches, but those experiments are not included here. The authors might want to clarify and perhaps test whether or not the measurements could be affected by the fluorescent label inherent to the approach.

We have clarified this important point, as described in the response to Reviewer #1 above. Measurements in ref. 47 were made using EMSA. The agreement (under the same conditions) by this approach and ³²P-RNA labelling (instead of protein labeling) suggests that the label has no impact as expected since it is far from the RNA binding site.

2) Affinities for 6,180 RNAs, while an impressive number, represent only a bit more than 2% of the entire sequence space of the 9-mer PUF4 binding site (> 262k sequence variants). The obtained model clearly describes the obtained affinities well and is carefully checked against overfitting (Suppl. Fig.4). However, since the data only cover a small sliver of the high affinity end of the binding spectrum, it might be premature to call this a “comprehensive” model. The vast

majority of the sequencing space of the 9-mer binding site is not included in the model, if I understand the model correctly. This terminology needs to be clarified. The described model is excellent for the high affinity part of the PUF4 affinity spectrum, but perhaps not “comprehensive”.

As the reviewer notes, our data are not “comprehensive”; however, our model is comprehensive, as it predicts the affinity for any RNA sequence. As we note a strength of this approach is that terms can be modified and additional terms added as called for by future data. We have clarified this distinction in the text between a comprehensive model vs. testing all possible sequences, which we cannot do. RNA-MaP is most powerful when guided by prior data that suggest the regions of sequence space to focus most intensively on, but it can also be used as a first pass discovery tool to obtain initial motifs and further hone from there.

New text in **bold & underlined**:

To derive a thermodynamic PUF4 binding model, we followed the approach we used previously with the related human Pumilio proteins, PUM1/2, designing a series of RNA sequences that vary relative to the previously-identified consensus sequence (Fig. 1a)⁹. In this case, the library was designed, rather than randomized, to ensure systematic variation relative to the consensus sequence, while also allowing exploration further into sequence space without sacrificing large amounts of the library to non-binders (Fig. 2a). **Other library approaches allow more sequences to be explored but sacrifice thermodynamic rigor. Also, the models from RNA-MaP, while not generated based on all possible sequences, are quantitative and predictive and can be expanded or modified if confronted by new affinities that, by rigorous statistical treatment, are not predicted by the current model.**

In our library we included mutations (1-4 nt) to probe sequence specificity and insertions (1-5 nt) to assay potential non-contiguous binding sites. Additionally, we varied the immediate flanking residues (0-3 nt) to assess possible extended binding sites (Fig. 2a and Supplementary Fig. 2a). The library included the sequence variants used to develop the PUM1/2 binding models²⁰ and variants from the consensus of PUF5 (an *S. cerevisiae* Pumilio protein related to PUF4)⁴⁴ to provide a range of related but different sequence variants (see Supplementary Table 1 for full library). The RNA sequence variants were embedded in four different scaffolds to control for RNA structures that could form from interactions with particular sequence variants or flanking sequences (Fig. 2a).

Reviewers' Comments:

Reviewer #1:

Remarks to the Author:

The authors made appropriate changes to the manuscript to address the suggestions I forwarded for their attention. The main issue was the importance of comparing to the greatest extent possible the specificity model for PUF protein binding to available binding data for specific sequence variants. I think this is a question that an audience with backgrounds outside the field of RNA binding protein specificity would be interested to know. The deeper discussion of previous validation of the RNA-MaP method and how it is more direct and well controlled than traditional biochemical methods is also important for a broad audience.

Alternatively, those familiar of RNA-protein interactions, who will appreciate the condition dependence of RNA binding protein affinity and apparent specificity, should find helpful the minor, but in my opinion important, additions to the text and Supp Fig. 3.

Reviewer #2:

Remarks to the Author:

The authors have addressed my concerns and I support publication

Minor comments:

Line 141 – I believe 'out library' is a typo ('our library')

Reviewer #3:

Remarks to the Author:

The authors have responded in a scholarly manner to the comments on the previous submission. I see no further major issues.

Two minor points, related to Figure 6.

1) The Figure legend is a narrative of the Figure content, instead of a description of the panels. That is unusual and - in my opinion - not helpful, because it requires the reader read and process the entire caption and the multiple fairly complex panels as a whole. In my estimate, a more traditional figure caption (separate descriptions of the panels, listed in order) is more beneficial for easy understanding.

2) Figure 6, panel g: The X-axis labels are hard to read.